# Chiral nanocrystals grown from MoS$_2$ nanosheets enable photothermally modulated enantioselective release of antimicrobial drugs

Bang Lin Li [1] ✉, Jun Jiang Luo[1], Hao Lin Zou[1], Qing-Meng Zhang[1], Liu-Bin Zhao [1], Hang Qian [2], Hong Qun Luo [1], David Tai Leong [3] ✉ & Nian Bing Li [1] ✉

The transfer of the concept of chirality from molecules to synthesized nanomaterials has attracted attention amongst multidisciplinary teams. Here we demonstrate heterogeneous nucleation and anisotropic accumulation of Au nanoparticles on multilayer MoS$_2$ planes to form chiroptically functional nanomaterials. Thiol amino acids with chiral conformations modulate asymmetric growth of gold nanoarchitectures on seeds of highly faceted Au/MoS$_2$ heterostructures. Consequently, dendritic plasmonic nanocrystals with partial chiral morphologies are synthesized. The chirality of dendritic nanocrystals inherited from cysteine molecules refers to the structural characteristics and includes specific recognition of enantiomeric molecules. With integration of the intrinsic photothermal properties and inherited enantioselective characteristics, dendritic Au/MoS$_2$ heterostructures exhibit chirality-dependent release of antimicrobial drugs from hydrogel substrates when activated by exogenous infrared irradiation. A three-in-one strategy involving synthesis of chiral dendritic heterostructures, enantioselective recognition, and controlled drug release system is presented, which improves nanomaterial synthetic technology and enhances our understanding of crucial chirality information.

Transferring chirality or handedness from light and enantiomeric molecules to synthetic materials, e. g. semiconductors, plasmonics, and tellurium nanocrystals, brings new life to nanotechnology in significant systems[1–4], enhances our understanding of biological and biomimetic processes and enables the development of practical applications[5–7]. Currently, large-scale and straightforward synthetic methods for nanosized chiral morphologies are still being developed[8]. Chiral nanostructures have stereochemical structures and likewise unique intrinsic properties, and remarkable effort has been expanded to study their functions and characteristics[9–11]. Among the burgeoning

materials, chiral gold nanostructures, which exhibit localized surface plasmon resonance (LSPR)-determined optical activities and enantioselective characteristics. They have aroused increasing multidisciplinary interest in aspects such as chemical synthesis, optical devices, and immunology[12]. For example, Kotov et al. reported that illumination of gold salt solutions with circularly polarized light (CPL) induced the formation of nanoparticles and subsequent assembly resulted in the construction of chiral nanostructures[13]. To support immunological studies, Xu et al. synthesized gold nanoparticles with controllable nanometer-scale chirality and high anisotropy by using

[1]Key Laboratory of Luminescence Analysis and Molecular Sensing, Ministry of Education, School of Chemistry and Chemical Engineering, Southwest University, Chongqing 400715, P. R. China. [2]Institute of Respiratory Diseases, Xinqiao Hospital, Third Military Medical University, 183 Xinqiao Street, Chongqing 400037, P. R. China. [3]Department of Chemical and Biomolecular Engineering, National University of Singapore, Singapore 117585, Singapore. ✉e-mail: chemlibl@swu.edu.cn; cheltwd@nus.edu.sg; linb@swu.edu.cn

CPL irradiation, and enantioselective approaches to immune cell regulation were announced[14].

Transferring chirality from biomolecules to plasmonic gold nanostructures gave birth to new members of the chiral family, which also exhibited enantioselective functionality. In terms of natural chiral molecular groups, enantiomeric amino acids have been used for the modulation of biological systems and play important roles in uncovering the mechanisms of life science[15]. Recognition and separation of L- and D-enantiomers of amino acids are essential in realizing scientific understanding and practical applications. The chiral amino acids imprinted the synthetic nanostructures, which in turn are equipped with distinguished chiral recognition basis for enantioselective release systems. As the chiral recognition capacities are very selective, the intrinsic characteristics of nanostructures are remarkably integrated in applications[16]. Nam et al. developed an amino-acid-directed synthetic route and proposed potential plasmonic metamaterial applications of the resulting chiral gold nanoparticles[17]. The growth of morphology-symmetric gold nanostructures from precursors of nanogold seeds were verified[17–19]. Based on the different interactions of L- or D- amino acids onto the gold crystal planes, sequential growth of the nanostructures on crystal seeds became anisotropic[20–22]. Significantly, the evolution of chirality in plasmonic nanostructures has contributed to biologically responsive and morphology-tunable metamaterials. Reaction strategies occurring at homogenous interfaces were beneficial in understanding the growth mechanisms and modulation roles of chiral molecules[17–19]. However, the atomic growth of gold at the heterostructural surfaces might cause morphological and functional differences, and those hybrids could exhibit a series of intriguing functions in many significant fields.

When using the two-dimensional (2D) nanostructures as the growth seeds, the resultant synthetic heterostructural products varied in morphologies and plasmonic and semiconducting properties[23–25]. Cysteine at the heterostructural surfaces caused anisotropic growth of the enantiometrically distinct nanomaterials. While there were a few reports of chiral structures of helical nanocrystals, this strategy has yet to be implemented with heterostructural nanocrystals exhibiting sizes of tens of nanometers. 2D layers are appeared to be comparable seeds and offer similar reaction sites to bring about the external transfer of chirality that could occur between amino acids and nanosized plasmonics. Furthermore, it is hoped that the intrinsic characteristics of 2D materials are integrated, and consequently, heterostructural conjugation of semiconductors and gold plasmonics will be exhibited[26–30]. Herein, we provide a comprehensive description of the formation of plasmonic gold nanostructures at MoS2 heterostructural interfaces. It was confirmed that when L- or D- amino acids were attached to the surfaces of the 2D layers, anisotropic growth of gold nanocrystals occurred at the heterostructural interfaces. Dendritic gold nanostructures with chiral characteristics are presented. The multilayer structures of 2D MoS2 were key factors in obtaining highly faceted nanocrystals, and the latter served as growth seeds, enabling the construction of dendritic nanoheterostructures. Due to the different interactions of the enantiomeric gold surface with antimicrobial drugs, a sustainable release system can be constructed with hydrogel and aerogel substrates. Based on the syntheses of cysteine-derived Au/MoS2 nanoarchitectures, development of chiral nanostructure-based antimicrobial strategies and infrared (IR) photothermally activated drug release was released

## Results

### Growth of chiral plasmonic nanocrystals

2D exfoliated MoS2 was obtained via sonication-induced liquid-phase exfoliation of MoS2 bulk crystals. Size separation and purification of the nanosheets were conducted with differential-speed centrifugation[31–33]. Ascorbic acid (AA) was added to HAuCl4 and hexadecyl trimethyl ammonium bromide (CTAB) solution to form the

CTAB-conjugated Au+ chemicals (CTAB-Au+)[17]. Upon addition of the as-synthesized exfoliated MoS2 nanosheets, the colorless CTAB-Au+ solution immediately changed to a blue color and finally turned into a purple dispersion (Supplementary Fig. 1). The time-resolved changes in the visual absorption spectra in the absence and presence of MoS2 were attributed to the sequential growth of the plasmonic nanostructures (Supplementary Fig. 2). Herein, the MoS2 nanosheets served as growth seeds, and the products collected from the reaction solution were identified as nanocrystals by scanning electron microscopy (SEM). Nevertheless, after the MoS2 seed solution was preincubated with micromolar L-cysteine (L-Cys), the colors of the L-Cys/MoS2 and CTAB-Au+ reaction mixtures gradually changed to blue. Optical absorption peaks showed redshifts from 564 to 877 nm with increasing concentrations of L-Cys (Fig. 1a, and Supplementary Fig. 2). The scattered blue light suggested that the MoS2 seeds and L-Cys resulted in morphological changes of the products from spherical nanocrystals to branched and even dendritic nanostructures[34–36]. The as-synthesized Au nanostructures moved through different configurations based on the different concentrations of L-Cys used to immobilize them on the MoS2 seeds (Fig. 1b). With a low concentration of L-Cys (0.4 μM), the products of cracked nanocrystals (CNCs) appeared to be few-facet polyhedrons, and the overall sizes of the particles remained constant (Fig. 1c). Subsequently, the number of facets increased with increasing L-Cys concentrations from 0.4 to 0.8 μM (Fig. 1b). When the concentration was gradually increased to 1.2 μM, facet surface growth began at the highly faceted nanocrystals with the construction of dendritic nanocrystals (DNCs-1). When the concentration range of L-Cys was increased to 1.6 and 2.0 μM, the partially branched areas of the synthesized nanostructures were even more accentuated, and the dendritic nanocrystalline products (DNCs-2 and DNCs-3), respectively, were formed. The maximum diameters of the nanostructures ranged from 90 to 130 nm, and the irregular sizes were attributed to the size diversity of the MoS2 seeds previously prepared via a top-down method from MoS2 powder (Supplementary Fig. S3)[31,32]. The energy dispersive X-ray spectra (EDX) indicated the element distributions in the CNCs and DNCs-3 groups. The distribution ratios of Au, S, and Mo elements were identical after doping of MoS2 into the Au nanostructures (Fig. 1d, e). Nevertheless, in terms of CNCs, the responses of Au elements at cracked holes were low, but that of the S element improved accordingly. The MoS2 seeds contributed to the formation of cracks on the Au nanocrystals due to exposure of Mo and S elements (Fig. 1d). The addition of micromolar cystine resulted in emerging branches on the Au/MoS2 nanocrystals, which exhibited no changes in elemental distribution. Similar synthetic experiments with nanocrystals were repeated with D-cysteine (D-Cys) instead of L-Cys. The chiralities of different nanostructures synthesized with the enantiomers of cysteine were studied by circular dichroism (CD) spectroscopy (Fig. 1f). The aqueous nanostructures exhibited two pairs of symmetric CD and g-factor profiles (Fig. 1f, g), indicating the handedness of the as-synthesized Au/MoS2 dendritic nanostructures. Due to the morphology-dependent LSPR characteristics of nanoplasmonics, the CD peaks at shorter wavelengths shifted from 550 nm to near 900 nm as the concentrations of L-Cys and D-Cys were increased from 0.4 to 2.0 μM[37–39]. After the concentrations of cysteine were increased to more than 1.6 μM, a pair of symmetric CD peaks appeared, and their intensities increased with increasing concentrations of L-Cys and D-Cys. When the cysteine concentration reached 2.4 μM, the symmetric CD peaks were located at consistent positive and negative wavelengths for the L-Cys and D-Cys samples, respectively. Remarkably, the shifts of CD peaks to long wavelengths were confirmed and are ascribed to the remarkable morphological diversity of symmetric nanoplamonics[14]. From 0.2 to 2.0 μM, the increasing concentrations of cysteine led to increases in the g-factor values from 0.002 to 0.008, which dropped to approximately 0.004 when the concentrations of cysteine enantiomers were increased to 2.4 μM (Fig. 1f).

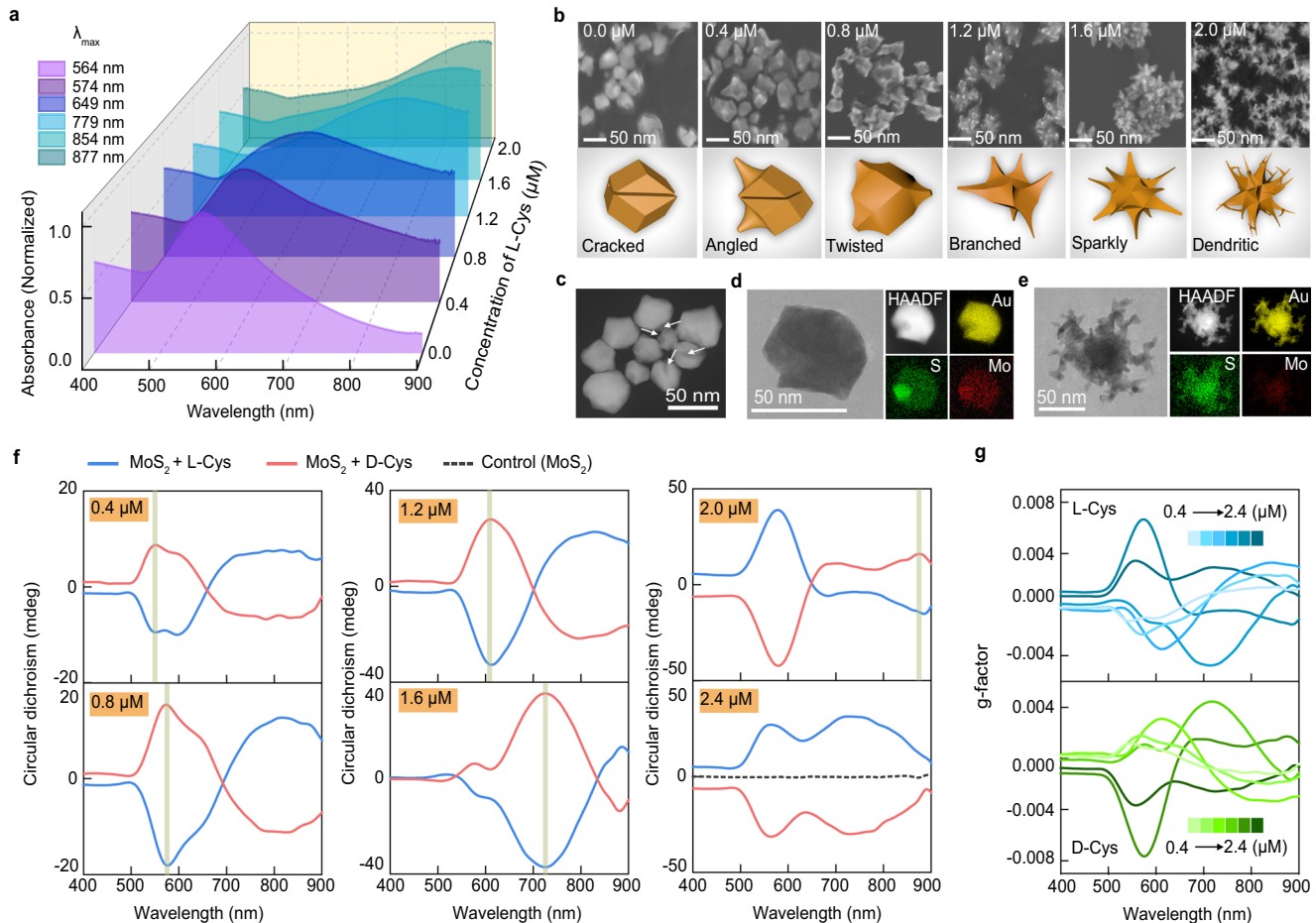

**Fig. 1 | Spectroscopic and morphological identification of hybrid nanostructures formed by the enantiomers of cysteine. a**, **b** Optical spectra (**a**), SEM images and corresponding simulated morphologies (**b**) of as-synthesized nanostructures based on growth with seeds of exfoliated MoS₂ nanosheets in the absence and presence of different concentrations of L-Cys (0.4–2.0 μM). **c**, **d** Dark-field TEM image (**c**), TEM and EDX element scan (**d**) of Au/MoS₂ nanostructures formed in the absence of L-Cys. **e** TEM and EDX element scans of Au/MoS₂ nanostructures synthesized with the addition of L-Cys (2.0 μM). **f**, **g** Circular dichroism spectra (**f**) and chiroptical *g*-factor responses (**g**) of Au/MoS₂ nanostructures synthesized in the absence (control MoS₂ sample) and presence of various concentrations of L-Cys and D-Cys (0.4–2.4 μM), respectively. Blue and red lines represent the circular dichroism curves from samples prepared from L-Cys and D-Cys, respectively. **a**, **f**, **g** Source data are provided as a Source Data file.

First, the vertical symmetry of the CD peaks indicated mirror symmetry of the crystal structures for the Au/MoS₂ nanostructures, which were related to the molecular symmetry of the cysteine enantiomers (Fig. 2a). SEM and TEM images exhibited consistent morphologies of DNCs-2 resulting from L-Cys (Fig. 2b, c) and D-Cys (Fig. 2d, e). In addition to the dendritic structures, nanostructures with "folded wings" were observed with TEM and the corresponding high-angle annular dark field (HAADF) image (Fig. 2f). The composition of exfoliated MoS₂ after reaction with CTAB-Au⁺ ions was identified and the elements Au, Mo, and S confirmed the formation of heterostructures comprising Au and MoS₂ hybrids (Fig. 2c and Supplementary Fig. 4). Morphologies of individual nanostructures of Au/MoS₂ DNCs-2 synthesized in the presence of 1.6 μM L-Cys and D-Cys were studied using diverse-angle TEM scans (Fig. 2g, h) and the corresponding TEM tomography images (Fig. 2i, j, Supplementary Movies 1, 2), respectively. Characterization results show that folded branches appeared on the nanocrystals, and the simulation indicated that the molecular chirality of L-Cys and D-Cys resulted in asymmetrical morphologies for the synthesized nanostructures (Fig. 2k). In previous studies, gold nanocrystals were employed as the growth seeds, and consequently, plasmonic helicoids, rather than dendritic structures, were fabricated[17,18]. The chiral branches of dendritic nanoplasmonics inherited handedness from the corresponding enantiomers of amino acids during the synthesis. We attribute this feature to the essential involvement of achiral MoS₂ heterostructural surfaces, whose growth mechanism was then studied in depth. The experimental results indicated that MoS₂ nanosheets worked as substrates and guided the synthesis of dendritic gold nanostructures.

## Mechanism for chiral transfer at molybdenum disulfide interfaces

Both MoS₂ heterostructural planes and thiol amino acids are essential for the formation of chiral gold nanostructures. In the presence of thiols, CNCs exhibited highly faceted structures and multilayer Au with attached MoS₂ nanosheets (Fig. 3a). We realized that Au/MoS₂ nanostructure growth was arrested by ligand exchange and solvent transfer. This simple step helped us to obtain time-resolved TEM images for the intermediate states of Au/MoS₂ nanostructures to investigate the role of MoS₂ in the growth mechanism. In terms of CNCs, the sizes of the Au crystals increased from 50 to 120 nm when the reaction time increased from 20 to 480 s (Fig. 3b). Cracks of CNCs were always found sandwiching the ultrathin MoS₂ nanosheets. In the presence of 1.2 μM L-Cys, CNC structures were formed within 120 s (Fig. 3c). HRTEM indicated that the thin layers near the nanocrystals were complete S-Mo-S hexatomic rings, and thus, the semiconducting crystalline material was identified[40]. After 180 s, branches were

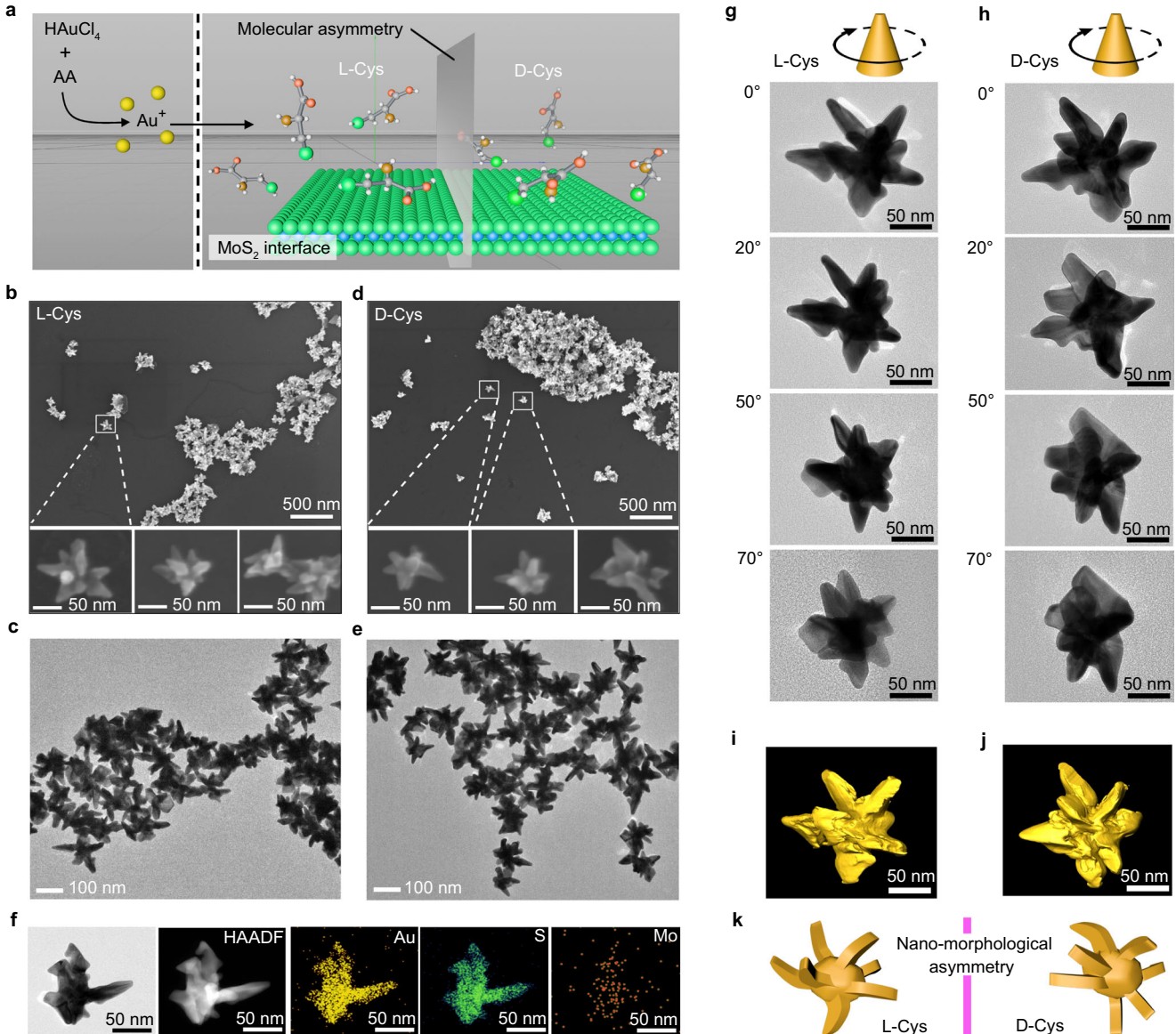

**Fig. 2 | Transfer of chirality from amino acids to gold nanocrystals. a** Schematic illustration of cysteine enantiomer-mediated growth of Au nanocrystals in the presence of exfoliated $MoS_2$ nanosheets; Mo and S atoms are shown as blue and green spheres, respectively. **b**–**e** SEM (**b**, **d**) and TEM (**c**, **e**) images of as-synthesized Au/$MoS_2$ nanostructures derived from 1.6 μM L-Cys (**b**, **c**) and D-Cys (**d**, **e**), respectively. **f** TEM, HAADF, and EDX element scan results of L-Cys-endowed branched Au/$MoS_2$ nanostructures. **g-j** Diverse-angle TEM scan (**g**, **h**) and corresponding TEM tomography images (**i**, **j**) for individual Au/$MoS_2$ nanostructures synthesized in the presence of 1.6 μM L-Cys (**g**, **i**) or D-Cys (**h**, **j**), respectively. **k** Asymmetric morphological simulation of synthesized Au/$MoS_2$ nanostructures based on TEM tomography images of nanostructures prepared with L-Cys (**i**) and D-Cys (**j**).

observed and finally progressed to dendritic nanoarchitectures with folded wings after 480 s. When the concentrations of L-Cys was increased to 1.6 μM, the cracked morphologies emerged much faster, within 20 s. Subsequently, dendritic structures with a few branches were rapidly formed (Fig. 3d). The particle sizes of DNCs-2 gradually increased, and they plateaued for 480 s (Fig. 3d). Therefore, the transition state (20 s) for DNCs-1 formation was studied (Fig. 3e, f). It was found that the thin nanosheets were embedded in the Au/$MoS_2$ nanocrystals, and both HAADF and EDX showed that the inserted thin nanosheets were layered $MoS_2$. Of significance, multiple Au layers are found during the growth of DNCs-1, which corresponded to the multilayer structures of exfoliated $MoS_2$ nanosheets (Fig. 3e, f). The overall growth of Au/$MoS_2$ composite nanostructures occurred along two paths (Fig. 3g)[41]. In the presence of $MoS_2$, Au atoms in the homogenous solution randomly attached to the edges and surfaces of exfoliated $MoS_2$ nanosheets (Fig. 3g). As edges have higher energies than $MoS_2$

surfaces, the Au deposition rate ($V_1$) was faster than that of surface deposition ($V_2$). Au atoms are preferentially attached to the $MoS_2$ edges. Moreover, when the atom surface diffusion ratio ($V_3$) was over $V_2$, Au growth was certainly dominated by surface diffusion towards the edges of the $MoS_2$ layers. Thus, after the edges were totally covered by Au atoms, heterogeneous nucleation of Au atoms spread onto the $MoS_2$ surfaces. As a control, the reaction was conducted in the absence of exfoliated $MoS_2$ nanosheets, and the planar surface and edge growth pathways merged and resulted in the formation of cuboctahedra (Fig. 3h)[42,43]. It is supposed that the cracks of highly faceted gold nanocrystals arose due to the variable thicknesses of the underlying exfoliated $MoS_2$ multilayers, as thicker nanosheets prevented the complete shrouding growth (Fig. 3i). Instead of cysteine, other thiol compounds, such as bovine serum albumin and glutathione, were also employed for the syntheses of Au/$MoS_2$ nanoheterostructures, and a few branches on the nanostructures were observed; their proportions

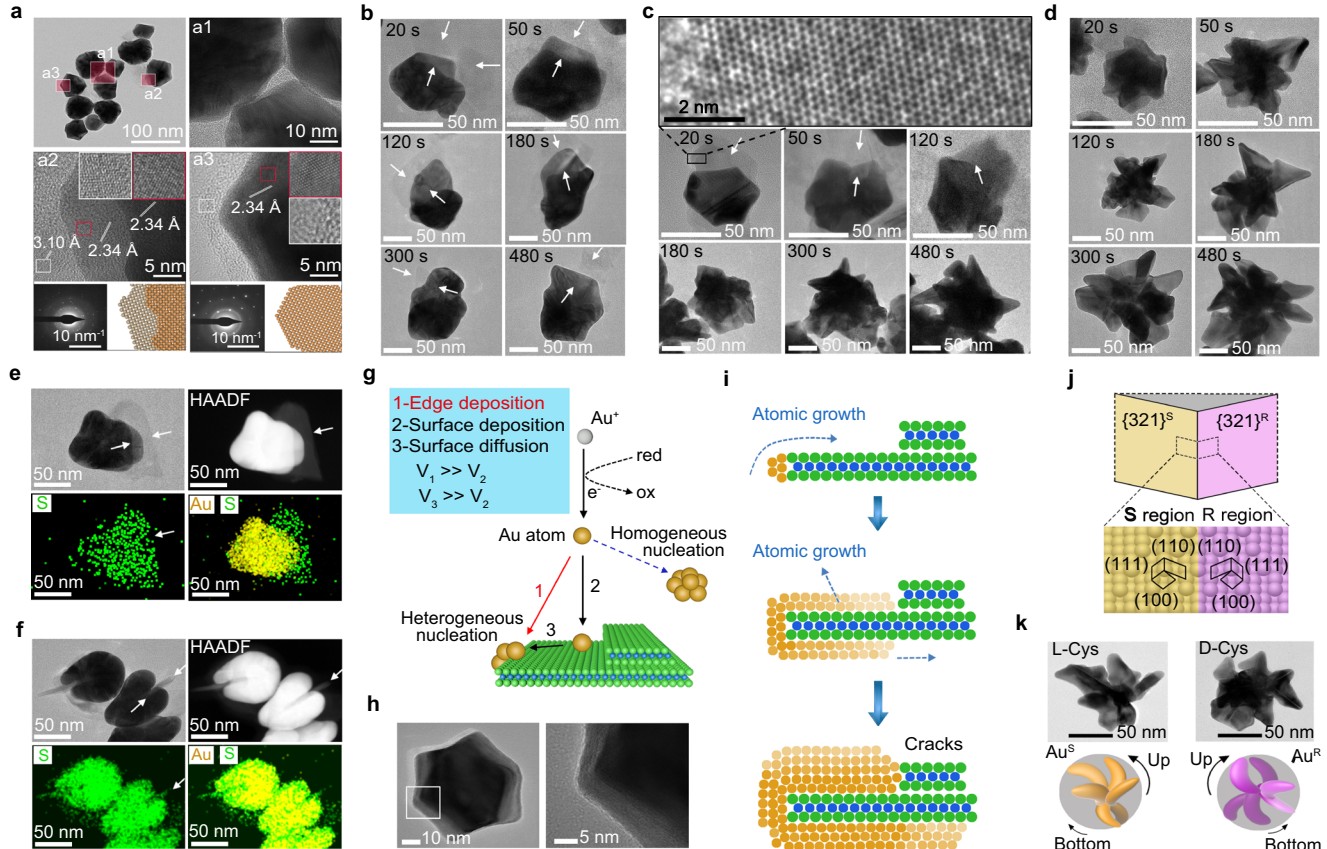

**Fig. 3 | Mechanism for anisotropic growth of Au nanocrystals on hetero-stuctural MoS$_2$ nanosheets. a** TEM, HRTEM, and the corresponding selected area electron diffraction pattern for Au/MoS$_2$ cracked nanocrystals (CNCs) which are synthesized in the absence of cysteine. **b**–**d** TEM and HRTEM images indicating time-resolved morphologies of Au/MoS$_2$ CNCs (**b**), DNCs-1 (**c**), and DNCs-2 (**d**), respectively. **e**, **f** TEM, HAADF, and EDX element map for Au/MoS$_2$ nanostructures obtained after reacting for 20 s in the presence of 1.2 μM L-Cys. **g** Illustration showing homogeneous nucleation of Au nanostructures in solution and hetero-geneous nucleation based on competitive deposition and diffusion of gold atoms on exfoliated MoS$_2$ nanosheets. **h** TEM and HRTEM images of crack-free Au nanocrystals synthesized by homogeneous nucleation in the reaction solution rather than at MoS$_2$ edges. **i** Anisotropic growth by deposition of Au atoms at edges of the multilayer MoS$_2$ nanosheets, resulting in cracked structures. **j** Schematic of R-S pairs showing morphological development and comparison of the atomic arrangements of the {321}$^R$ and {321}$^S$ gold surfaces for region indicated by the dotted box. **k** TEM images and schematics of DNCs-1 synthesized in the presence of L-Cys and D-Cys, respectively.

were lower than those resulting from the cysteine synthesis (Supplementary Fig. 5). The thiol groups in L-Cys and D-Cys are critical in forming dendritic growth, and amino acids that lacked thiol groups, e.g. L-glutamic acid and L-arginine, did not induce dendritic growth (Supplementary Fig. 6). Furthermore, thiol compounds, such as mercapto-ethylamine, still led to dendritic nanostructures with morphologies similar to those of nanocrystals formed by the thiol amino acids (Supplementary Fig. 6). Therefore, coexisting thiol compounds have contributory effects in the formation of dendritic structures when gold nanostructures were grown in-situ on MoS$_2$ layers.

In the presence of cysteine molecules, the intermediate states of highly faceted nanostructures guide the formation of Au/MoS$_2$ nano-heterostructures with dendritic shapes. To support the chirality transfer mechanism, the synthetic routes were changed, and CD spectra of the final products were measured (Supplementary Fig. 7). The dispersed cysteine molecules were removed from the MoS$_2$ seed solution via centrifugation. Consequently, the chiral Au/MoS$_2$ products were not synthesized, which indicated that the chirality of the nanoheterostructures originated from cysteine enantiomers in solution, instead of molecules physisorbed on MoS$_2$ surfaces. Exfoliated MoS$_2$ was first added to the growth solution (CTAB-Au$^+$) as a negative control. After different reaction times, the L/D-enantiomeric cysteine solutions were added to the reaction solutions, respectively. CD analyses of products (Supplementary Fig. 7) indicated that the chirality of

the DNCs was derived from the selective interactions between cysteine enantiomers and reaction intermediates with highly multifaceted Au/MoS$_2$ nanoheterostructures, instead of interaction differences between cysteine enantiomers and MoS$_2$ surfaces. The syntheses of chiral structures arose from the different growth rates for the two enantiomers of high-index planes of gold in the presence of L-Cys or D-Cys[17,18]. Growth on the highly multifaceted Au polyhedron nanostructures depends on L/D-Cys templating on those {321} facets (Fig. 3j). The {321} facets exhibited the R (clockwise rotation, {321}$^R$) or S (anticlockwise rotation, {321}$^S$) conformation, which was defined by the rotational direction of the low-index planes [100], [110] and [111], as outlined black in Fig. 2i. The R and S triangular regions alternated and their distributions were symmetric. The chiral morphologies were attributed to the shifting and tilting of specific R-S boundaries. There were many facets, indicating the R-S boundary on the multilayer Au/MoS$_2$ nanocrystals. The orientation of the cysteine molecules adsorbed on the surfaces of seeds determined the specific growth direction. The thiol and amine groups bind at "*kinks*" on the {321} facets. The preferential interaction of L-Cys with the {321} planes in the R regions led to slower vertical growth in the R regions than that in the S regions[17,44,45]. Thus, the R-S boundary shifted from the R to the S region, which was accompanied by the asymmetric overgrowth of gold nanostructures. As a consequence, the branch structures of DNCs were constructed. The overgrowth S regions seen when L-Cys was used were

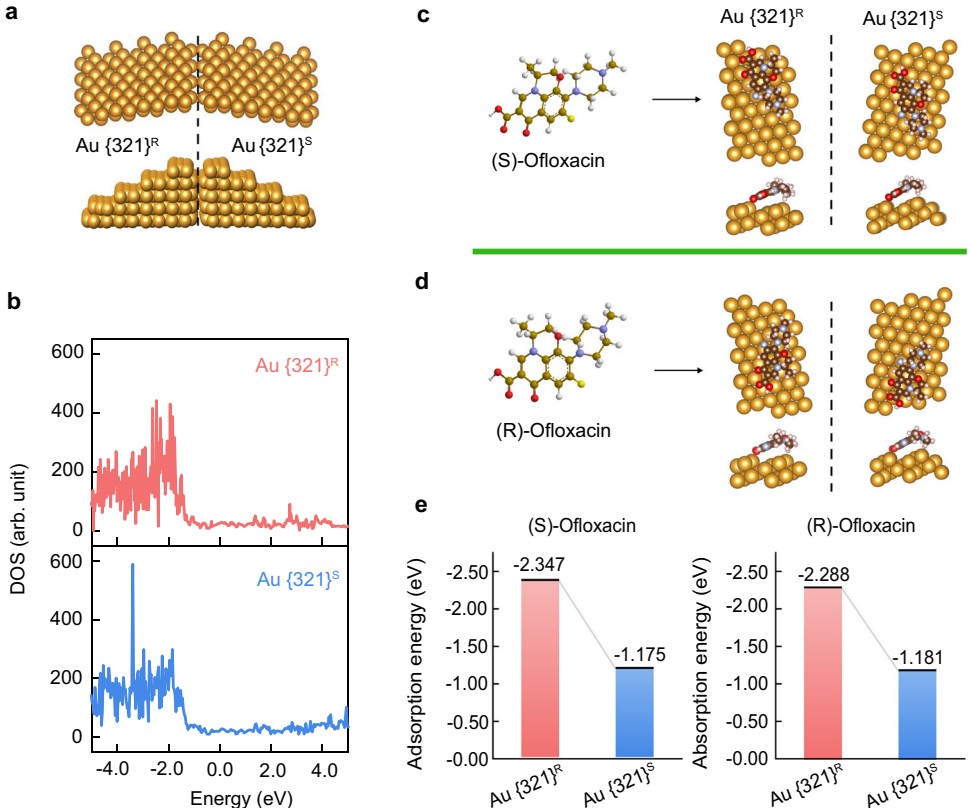

**Fig. 4 | Simulation indicating different interactions between ofloxacin enantiomers and chiral gold surfaces. a** Comparison of the atomic arrangement of $\{321\}^R$ and $\{321\}^S$ gold surfaces. **b** Density of state (DOS) results show the variety of structures or chiral crystalline faces. **c, d** Optimized adsorption modules of (S)/(R)-ofloxacin molecules on the chiral crystalline faces. **e** Adsorption energies of (S)/(R)-ofloxacin molecules interacting with various chiral $\{321\}^R$ and $\{321\}^S$ gold crystal surfaces. **b** Source data are provided as a Source Data file.

more pronounced than those produced when D-Cys was used. Herein, L-Cys-endowed DNCs are labeled as (S)-DNCs nanostructures. Conversely, the same logic holds true for asymmetric DNCs induced by D-Cys, which had overgrown R regions, and the products were consequently designed as (R)-DNCs nanostructures. The chiral nanocrystals of (S)-DNCs-1 and (R)-DNCs-1 were identified by the TEM studies of the L-Cys and D-Cys-related nanoheterostructures (Fig. 3k), respectively. A slight zigzag was observed for each branch of DNCs-1, and those zigzags were individually asymmetric on the L-Cys- and D-Cys-derived Au/MoS$_2$ nanoheterostructures.

### Discriminated interaction effects of chiral crystal surfaces

The different effects of the enantiomeric gold facets on molecular interactions were investigated by theoretical simulations[46,47]. First, the R and S triangular regions alternated and their distribution was essentially symmetric (Fig. 4a). The density of states (DOS) data were analysed, which indicated the different electron distributions of the enantiomeric nanostructures (Fig. 4b). When the chiral molecules were attached on the R- or S-rich nanostructures, the adsorption modules varied due to enantiomeric differentiation. Using this logic, we explored the possibility of specific capture and release of chiral molecules using enantiomeric surfaces. Using chiral ofloxacin molecules, enantiomeric gold crystal surfaces with the lowest energies were presented in terms of (S)-ofloxacin (Fig. 4c) and (R)-ofloxacin (Fig. 4d). The theoretical simulations indicated that ofloxacin molecules, no matter what chirality they are, have lower adsorption energy on the $\{321\}^R$ gold surface (Fig. 4e), and the $\{321\}^R$-rich nanostructures were preferentially and physically bound with the ofloxacin drugs. With integration of the intrinsic characteristics of plasmonics and the emerging properties of dendritic nanoheterostructures, chiral

nanocrystals exhibit huge potential for developing responsive and sustainable release systems.

### Photothermally responsive and enantioselective hydrogels

The gold and MoS$_2$ nanostructures exhibited comparable photothermal and biological compatibilities[47,48]. Photothermally-responsive drug release was synergistically motivated by the hybrid of gold and MoS$_2$ materials. The chirality was transformed from amino acids to inorganic heterostructures with dendritic morphologies. Additionally, it is supposed that the chiral interaction functions of amino acids in nature was also transferred. Therefore, the characteristics emerging from the inherited chirality and integrated properties of the heterostructures were studied. Due to synergistic effects, dendritic nanoheterostructures exhibit intriguing potential for use in biological applications. Hydrogels are considered comparable carriers of nanostructures, and they can support practical applications of as-prepared dendritic nanoheterostructures[49,50]. DNCs were mixed in a warmed agarose aqueous solution, and the mixture was left to cool to form DNCs doped agarose hydrogels. The hydrogel samples were freeze-dried in order to obtain composite DNCs-doped aerogels (Fig. 5a). The UV-Vis-NIR absorption spectra of CNCs, DNCs-1, and DNCs-2 hydrogels were recorded. In comparison to CNCs, which only exhibited optical absorptions in the visible range, (S)/(R)-DNCs exhibited strong absorption in the near-infrared (NIR) range (Fig. 5b). Doping of Au/MoS$_2$ nanoheterostructures into both agarose hydrogels and aerogels enhanced the photothermal characteristics. The DNCs hydrogels exhibited higher photothermally-induced temperature increases than aerogels, which was attributed to efficient heat transmission by water in the hydrogels (Fig. 5c). The DNCs with differential configurations even exhibited consistent NIR absorption and

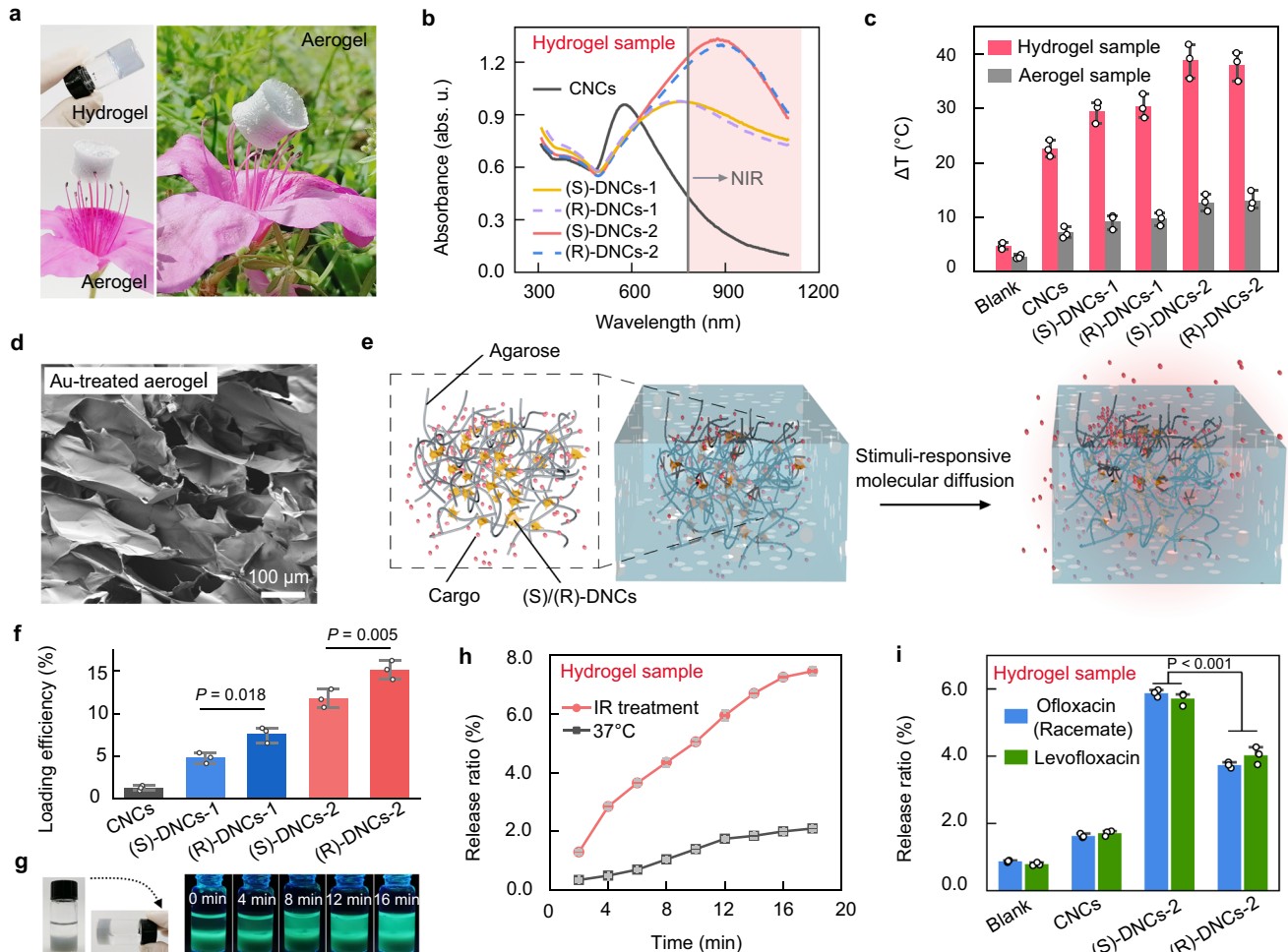

**Fig. 5 | IR-responsive drug release properties of hydrogels modified by chiral dendritic nanocrystals. a** Photos of agarose hydrogels and an aerogel doped with nanocrystals. **b** UV-vis-NIR spectra showing the different optical absorption spectra of the cracked nanocrystals (CNCs), DNCs-1, and DNCs-2 hydrogels. **c** Temperature changes of aerogels and hydrogels doped with various nanostructures before and after IR (808 nm) treatments. **d** SEM image of the hybrid aerogel (0.8% agarose) after pretreatment with gold spraying. **e** Schematic illustration showing the composition of the hybrid hydrogel packed with cargo and release of the cargo from the hybrid hydrogel based on molecular diffusion. **f** Saturation loads of levofloxacin for diverse Au/MoS$_2$ nanostructures. $P < 0.05$; the significant difference in the levofloxacin loading efficiencies for (S)-DNCs and (R)-DNCs nanostructures. **g** Photos of levofloxacin-loaded DNCs-2 hydrogels covered by aqueous solution in the upper layer and UV-irradiated levofloxacin-loaded DNCs-2 hydrogels after various times

for the IR treatments. **h** Drug release ratios from levofloxacin-loaded DNCs-2 hydrogels to the upper-layer aqueous solutions under IR treatment (red curve), which were compared with incubation at 37 °C (black curve). The loading amount of levofloxacin was 50.00 µg mL$^{-1}$. **i** Release ratios for levofloxacin (green bars) and racemic ofloxacin (blue bars) on a blank agarose hydrogel and different Au/MoS$_2$ nanostructure-doped agarose hydrogels after IR treatment for 10 min. Hydrogels were preloaded with levofloxacin and ofloxacin (50.00 µg mL$^{-1}$), respectively. $P < 0.001$; significant difference of (R)-DNCs-2 and (S)-DNCs-2 doped hydrogels in ofloxacin racemate and levofloxacin release ratios. **c, f, h, l** Error bars representing the standard deviations (SDs) of independent experiments ($n = 3$). Data are presented as mean values ± SDs. **f, i** Statistical analyes were performed via one-way ANOVA followed by Tukey's multiple comparison tests. **b, c, f, h,** and **i** Source data are provided as a Source Data file.

photothermal properties, and after IR irradiation (3.0 W/cm$^2$) for 5 min, the temperatures of the CNCs, DNCs-1, DNCs-2 hydrogels increase from 25 °C to ~47, 55, and 65 °C, respectively (Fig. 5c). Significantly, efficient photothermal conversion of DNCs was attributed to the strong plasmon resonance absorption of the infrared laser (808 nm). Among the Au/MoS$_2$ nanostructures studied, DNCs-2 exhibited the highest photothermal conversion efficiency. Due to the branch-rich structures, the chiral gold surfaces of DNCs-2 exhibited different interactions with molecular enantiomers. DNCs-2 was thus considered for further hydrogel-based biomedical applications. Networked porous channels within the hydrogels exhibiting pore sizes ranging from 2 to 100 µm were prepared with varying agarose concentrations (Fig. 5d and Supplementary Fig. 8). These channels allowed free diffusion of small molecules. Therefore, to construct a stimuli-responsive release system (Fig. 5e), the drugs were loaded into the hydrogels via the

addition of levofloxacin to the heated (S)/(R)-DNCs-doped agarose dispersion.

Levofloxacin exhibited strong green emission under the UV irradiation, and its fluorescence characteristics were used for the quantitative concentration analyses (Supplementary Fig. 9). To assess the different interactions between ofloxacin and the chiral nanostructures in aqueous solution, levofloxacin and ofloxacin racemates were incubated with (S)-DNCs-2 or (R)-DNCs-2, respectively, for different times (Supplementary Fig. 10). Subsequently, the aqueous systems were centrifuged, and the fluorescence intensities of levofloxacin and ofloxacin in the supernatant were measured. The fluorescence intensities decreased with increasing incubation time, indicating rapid adsorption of drug molecules on the plasmonic nanomaterials. More importantly, in terms of (S)-DNCs-2, the fluorescence changes of ofloxacin were larger than those of levofloxacin. In contrast, (R)-DNCs-2 exhibited higher fluorescence changes to levofloxacin. The different

capacities of (S)-DNCs-2 and (R)-DNCs-2 for the adsorption of ofloxacin enantiomers were identified. As a control, no differences were found for the CNCs. Time-resolved fluorescence data indicated that the loading of the drug molecules on the DNCs was rapid and that the adsorption of ofloxacin was completed within 30 mins. In addition, we found that the saturated loading efficiency of levofloxacin on (R)-DNCs was higher than that on (S)-DNCs (Fig. 5f). R enantiomers of DNCs-1 exhibited higher loading efficiency than S enantiomers, which was attributed to the stronger interaction of {321}$^R$ gold surfaces with (S)-ofloxacin. A similar trend was found for the DNCs-2 nanostructures. In addition, due to the heavily branched structure, (R)-DNCs-2 had a larger {321}$^R$ gold area, which resulted in a larger efficiency for ofloxacin loading than (R)-DNCs-1. To check for the in-vitro drug release kinetics, the buffer solution was first added to the levofloxacin-loaded DNCs hydrogels, which allowed diffusion of free levofloxacin from the hydrogels to the buffer solution. The concentrations of levofloxacin were measured in the buffer solution to assess the release ratios of the loaded hydrogels over time (Fig. 5g). With increasing incubation time, the fluorescence intensities of the buffer solutions increased accordingly. IR treatment enhanced the diffusion of levofloxacin from the hydrogel to the aqueous phase, and the levofloxacin increase ratio rose from approximately 2.1% to approximately 7.8% (Fig. 5h). In comparison to hydrogel samples without DNCs, DNCs-doped hydrogels exhibited IR-responsive characteristics, which suggested that heat accelerated the diffusion of molecular drugs, and consequently, exogenous IR-responsive drug release systems have been developed.

The IR-responsive drug release system was studied (Fig. 5i). The concentrations of levofloxacin and ofloxacin release by CNCs and DNCs-2 doped hydrogels were analysed. After IR treatment, the release ratios of the DNCs-2 hydrogel were higher than those of the CNCs. There was a slight difference between the ofloxacin release ratios of (S)-DNCs-2 and (R)-DNCs-2. It is supposed that ofloxacin diffusion in (R)-DNCs was slower than that in (S)-DNCs. In comparison, we found that (S)-DNCs-2 exhibited higher release ratios for levofloxacin, which reached approximately 6% after IR treatment for 20 min. In contrast, (R)-DNCs-2 exhibited enhanced release of ofloxacin, which had larger release ratios than levofloxacin. This result indicated that the chiralities of Au-MoS$_2$ nanoheterostructures produced differernt adsorption levels for chiral drugs. The cores of the hydrogels enabled the free diffusion of small molecules from the hydrogel to the environment, and consequently, the differences caused by recognition of chiral drugs by the DNCs reduced. To confirm the value of practical application, a chirality-dependent drug release system was developed from Au/MoS$_2$ DNCs-2 enantiomers. In comparison to (R)-DNCs-2, (S)-DNCs-2 had a much stronger interaction energy with levofloxacin, which generated efficient antimicrobial effects. This finding contributed to establishment of a controllable and sustained release formulation[51–55].

## Responsive and sustained antimicrobial system

Nanostructure-based combination therapies show promise in combating bacterial infections and have the capacity to evade existing mechanisms for acquired drug resistance. Large-dose antimicrobials have exhibited unstable chemical properties, limited antibacterial, and detrimental side effects, and collectively promoted drug resistance. Consequently, the advanced intelligent drug delivery system was studied in the hydrogel phase[54–59]. In terms of antimicrobial assessment, co-incubation strategies indicated that the growth of *S. aureus* was largely restrained in the presence of dendritic nanostructures, and both (S) and (R)-enantiomers exhibited comparable antimicrobial activities due to the destruction of the bacterial cell membrane by reactive oxygen species (Supplementary Fig. 11)[60,61]. The doped hydrogels were made into tablets, which were preloaded with levofloxacin (Fig. 6a). The fluorescence of levofloxacin-loaded (R)-DNCs-2 and (S)-DNCs-2 agarose hydrogels (AGHs) was significantly weaker than that of the control AGH loaded with levofloxacin (Fig. 6b). Au/

MoS$_2$ heterostructures efficiently quenched the fluorescence of levofloxacin, and the fluorescence changes were employed to track the drug. Antimicrobial research with *S. aureus*, *E. coli*, and *P. aeruginosa* bacteriostatic circles (BCs) showed that the DNCs-doped hydrogel tablets exhibited weak antimicrobial efficiencies, and were enantiomerically independent (Fig. 6c–e). When the tablets were loaded with levofloxacin, the BC values increased with increasing concentrations of levofloxacin. In contrast, when the concentrations of the nanostructures increased, the BCs of the hydrogel tablets decreased correspondingly. However, the BCs formed by control AGHs were still larger than those of the (S)-DNCs/(R)-DNCs-2 AGHs, whereas (R)-DNCs-2 exhibited the smallest BC (Fig. 6f). Release of levofloxacin from the (R)-DNCs-2-doped hydrogel was the slowest among the three samples, which was supported by our simulation result, indicating that (R)-DNCs-2 had higher interaction energy with levofloxacin molecules than (S)-DNCs-2. With increasing concentrations of levofloxacin, adsorption by the nanostructures was saturated, and the BC differences for the DNCs AGHs and the control AGHs gradually decreased (Fig. 6f). Sustainable drug release system by hydrogel tablets could be developed with different interactions between the enantiomeric dendritic nanostructures and levofloxacin. One treatment period refers to the incubation of AGHs on top of the bacterial culture medium. During the first period (1st Period), the blank AGH sample exhibited antimicrobial performance comparable to that of the (S)-DNCs/(R)-DNCs-2 AGHs (Fig. 6g, h, and Supplementary Fig. 12). The BC diameters dropped substantially to approximately 12 mm during the second period (2nd Period), and exhibited no antimicrobial characteristics after the third treatment period (3rd Period). In contrast, the antimicrobial effect of levofloxacin on DNCs-2 AGHs was retained in the 3rd Period, and it continued even into the fourth period (4th Period). More importantly, even though the BCs of (R)-DNCs-2 AGHs were smaller than those of (S)-DNCs-2 AGHs in the 1st Period and 2nd Period, the sustainable operations of (R)-DNCs-2 AGHs were better, indicated by the BCs formed in the 3rd Period and 4th Period (Fig. 6h). Significantly, comparable properties for sustained drug release were exhibited by the dendritic Au/MoS$_2$ nanoarchitectures. Additionally, the photothermal characteristics of heterostructures can significantly enhance the activities of responsive antimicrobial systems. The photothermal and IR-activated antimicrobial activities of the tablets were studied. With IR irradiation (1 min), the temperatures of the DNCs-doped tablets increased rapidly to over 50 °C (Fig. 6i). Improvements in photothermal performance were achieved with increasing concentrations of the doped Au/MoS$_2$ nanostructures. In the 1st Period, the experimental results showed that IR irradiation had no impact on the BCs of levofloxacin-free AGHs or the levofloxacin-free (S)-DNCs-2 and (R)-DNCs-2 AGHs. After the hydrogels were loaded with levofloxacin, the BCs largely increased for the (S)-DNCs-2, and (R)-DNCs-2 doped hydrogel samples, but the BCs remained constant for the blank AGHs (Fig. 6j). These differences were attributed to accelerated molecular diffusion induced by the photothermal activation, and thus, the amount of levofloxacin released from the dendritic nanoheterostructures increased. The IR-activation strategy also works in sustainable antimicrobial experiments. In terms of the 2nd Period, 3rd Period, and 4th Period, the IR activation capacities of (R)-DNCs-2 hydrogels were better than those of (S)-DNCs-2 hydrogels (Fig. 6k). Due to the integration of the levofloxacin-DNCs enantioselective interactions and photothermal characteristics, advanced synergistic IR-activated and sustainable drug release systems based on the chiral dendritic nanoarchitecture-doped hydrogels and aerogels have a high potential for development of intriguing commercial products (Fig. 6l).

## Discussion

Liquid-exfoliated MoS$_2$ nanosheets with multiple layers were employed to seed the growth of chiral Au nanostructures and form Au/MoS$_2$ nanoheterostructures. The different interactions of chiral

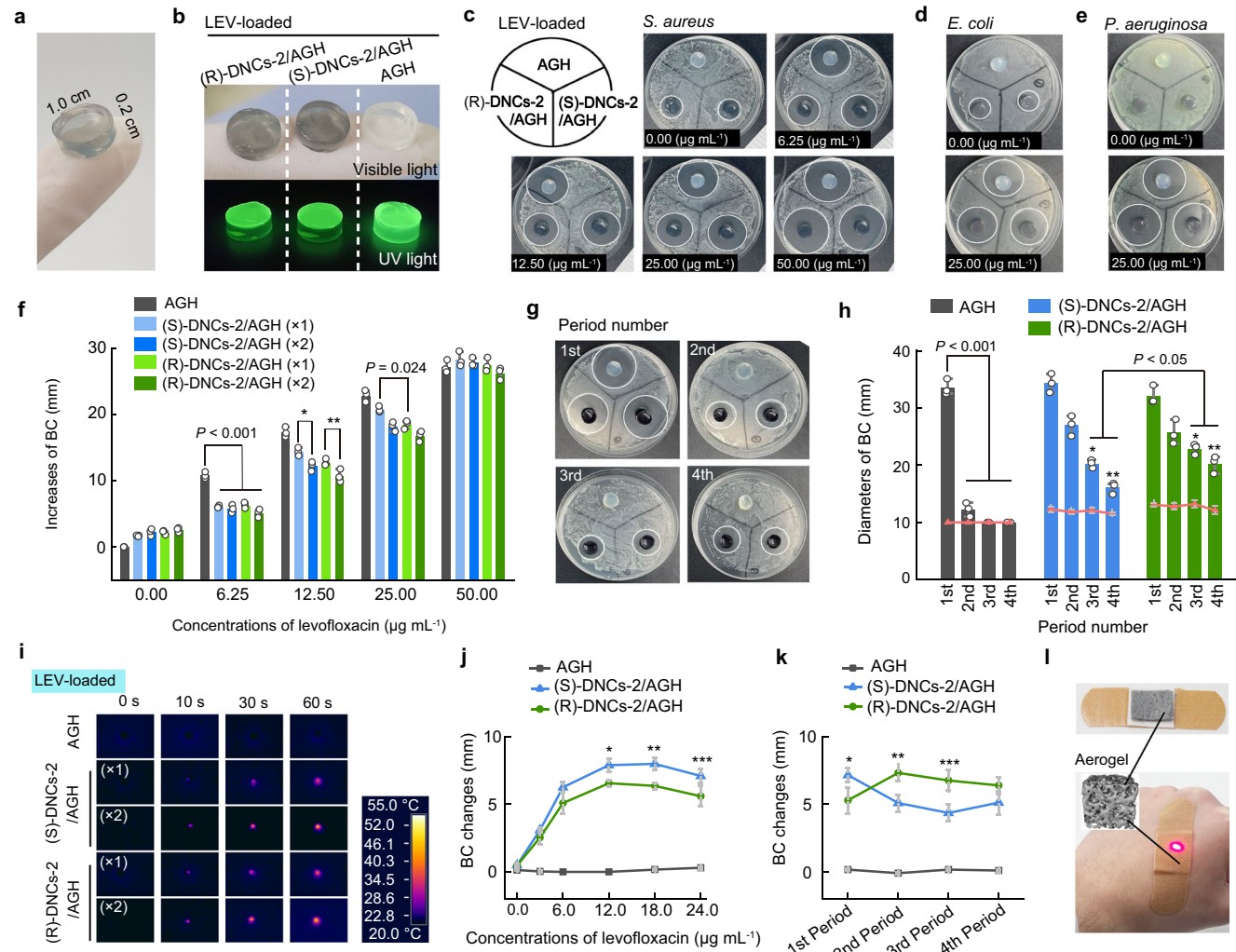

**Fig. 6 | Enantioselective, IR-activated, and sustainable antimicrobial hydrogels.**
**a** Photograph of a nanostructure-doped hydrogel tablet. **b** Photos showing
levofloxacin-loaded hydrogel tablets without and with doping of (R)-DNCs-2 and
(S)-DNCs-2 under irradiation with visible and UV light. **c**–**e** Photos indicating the
bacteriostatic circles (BC) of *S. aureus* (**c**), *E. coli* (**d**), and *P. aeruginosa* (**e**) for tablets
loaded with various concentrations of levofloxacin (**c** 0.00-50.00 μg mL⁻¹; **d**, **e** 0.00
and 25.00 μg mL⁻¹). **f** Statistical BC results for different tablets loaded with various
concentrations of levofloxacin. The concentrations of the nanostructures were
100 μg mL⁻¹ (×1) and 200 μg mL⁻¹ (×2). *$P$ = 0.035, **$P$ = 0.024; the significant differ-
ence between BC increases of levofloxacin-loaded (S)-DNCs-2/AGH and (R)-DNCs-2/
AGH. **g**, **h** Pictures (**g**) and statistical results (**h**) indicating sustainable release per-
formance based on the period-dependent BCs of diverse levofloxacin-loaded
hydrogel tablets. The red curves in **h** show the BCs of hydrogel tablets without
levofloxacin. *$P$ = 0.008, **$P$ = 0.007; significant difference between levofloxacin-
loaded (S)-DNCs-2/AGH and (R)-DNCs-2/AGH in the 3rd and 4th periods of sus-
tained antimicrobial experiments. **i** Diverse tablets irradiated with IR light for a

period of 60 s. **j** BC changes in AGH, (S)-DNCs-2/AGH, and (R)-DNCs-2/AGH tablets
before and after IR activation. The tablets were loaded with different concentra-
tions of levofloxacin. *$P$ = 0.004, **$P$ = 0.001, ***$P$ = 0.032; the significant difference
between IR-induced BC changes for levofloxacin-loaded (S)-DNCs-2/AGH and (R)-
DNCs-2/AGH. **k** BC changes indicating IR-activation improved the drug release
performance in sustainable antimicrobial applications. The concentration of levo-
floxacin was 18.0 μg mL⁻¹. *$P$ = 0.024, **$P$ = 0.004, ***$P$ = 0.005; the significant differ-
ence between IR-induced BC changes for levofloxacin-loaded (S)-DNCs-2/AGH and
(R)-DNCs-2/AGH in the 1st, 2nd, and 3rd periods of sustained antimicrobial
experiments. **l** Scheme showing wound repair using levofloxacin-loaded (R)-DNCs-
2 agarose aerogels with IR-activated and sustainable release. **f**, **h**, **j**, **k** Error bars
represent the SDs of independent experiments (n = 3). Data are presented as the
mean values ± SDs. Statistical analysis was performed through one-way ANOVA
followed by Tukey's multiple comparison tests. Source data are provided as a
Source Data file.

cysteine molecules with the {321}ᴿ and {321}ˢ crystal planes of Au
nanocrystals on the layered MoS₂ surfaces resulted in sequential ani-
sotropic growth of nanomaterials at the heterostructure interfaces. A
protocol for the syntheses of morphology-diverse nanohetero-
structures was presented, and helical and branched structures of plas-
monic gold nanocrystals were modulated with the concentrations of
amino acids to produce nano-stars, branched nanocrystals, and den-
dritic morphologies with chiral nanostructures. Chirality transfer from
the biomolecules of thiol amino acids to Au/MoS₂ nanohetero-
structures was identified. In addition, integrated photothermal reac-
tions and the chiral characteristics of the as-synthesized dendritic

materials were confirmed. The capacity for chiral recognition of
enantiomeric organic molecules was transferred from amino acids to
the dendritic nanoarchitectures. The different interactions of enan-
tioselective dendritic nanoarchitectures with levofloxacin enabled
modulated drug release. The chiral characteristics of the nanocrystals
were integrated with their intrinsic photothermal properties, and IR
photonic-activating enantioselective and sustainable release systems
for chiral drugs were developed based on the substrates of network
porous hydrogels. The three-in-one route brings more possibilities of
chirality for dendritic nanocrystals, emerging chiral characteristics,
and intriguing functions. In addition, the performance of the chiral

nanostructures can be further improved by the screening of hydrogel substrates. The synergistic characteristics of substrates will inspire applications of the emerging chiral plasmonic nanomaterials for the construction of optoelectronic devices and the development of important biological information.

## Methods

### Chemicals and materials

Gold (III) chloride trihydrate (HAuCl$_4$·3H$_2$O), was purchased from Adamas-beta Chem. Co., Ltd. (China). Levofloxacin (98%), L-cysteine (99.8%, BBI), and ofloxacin (USP grade) were obtained from Aladdin Chem. Co. Ltd. (Shanghai, China). D-Cysteine (99%) was obtained from Acros Organics (United Kingdom). Agarose (for electrophoresis use grade) was purchased from BBI (Sangon Biotech Co., Ltd., Shanghai, China). The thiolated polystyrene reagent was synthesized by Xi'an ruixi Biological Technology Co., Ltd. (Xian, China). All chemicals were used as received without further purification. *S. aureus* (CMCC(B) 26003), and *E. coli* (CMCC44102) were obtained from the Chongqing Boer Biotech. Co. Ltd. (Chongqing, China). *P. aeruginosa* (BNCC360090) was obtained from BeNa Culture Collection (Zhengzhou, China).

### Characterization and instruments

Five hundred microlitres of nanocrystals (suspended in 1 mM CTAB solution) were washed twice with deionized water to produce nanoparticles suspended in 5 μM CTAB solution. As-prepared nanocrystal dispersions were diluted with ethanol, and then a portion of the 50 μL mixture was dropped onto the silicon substrate and dried under vacuum (50 °C). The silicon substrates coated with plasmonic nanoparticles were washed with deionized water and dried by blowing with N$_2$ gas. The photonic absorption spectra were recorded on a UV-2450 spectrophotometer (Shimadzu, Japan), and the UV-vis-NIR spectra scan was performed with a UH-4150 spectrophotometer (Hitachi, Japan). CD spectra were obtained using a MOS-500 spectropolarimeter instrument (Bio-Logic, French) and a J-810 CD spectrometer (JASCO Corporation, Japan). HRTEM and 3D tomography tests were conducted with a Talos F200X instrument (FEI, USA). Purification of nanomaterials was achieved with centrifugation (Multifuge X1R, Thermo Scientific). Drug release was achieved via oscillation (Orbital Shaker TS-1000, Kylin-Bell Lab Instruments). SEM results were collected with a SU8220 system (Hitachi, Japan). An infrared thermal imager (E4 FLIR, USA) was used for photothermal imaging. Temperatures were controlled with a heating magnetic stirrer (C-MAG HS7, IKA). Exfoliation of the MoS$_2$ powders was performed in a bath sonicator (KQ-400B, Kunshan Instrument Co. Ltd., China). The circulating-water temperature-control system was provided by Zhengzhou Great Wall Scientific Co. Ltd. (HX-1005, China).

### Liquid-phase exfoliation of MoS$_2$ crystals

A mixed dispersion containing 5.0 mg mL$^{-1}$ MoS$_2$ crystal powders and 1.5 mg mL$^{-1}$ sodium cholate was treated with the bath sonicator for 20 h with a circulating-water temperature-control system (15 °C). The black powdered dispersion turned dark green after sonication. To remove the unexfoliated bulk crystals and thick sheets, a portion of the 1.50 mL treated dispersions was centrifuged at a relative centrifugal force of 987 × $g$ for 30 min, and subsequently, the yellow-green supernatant (1.30 mL) was collected. Furthermore, the collected dispersion of MoS$_2$ layers was purified. First, it was treated with high-speed centrifugation (10965 × $g$) for 30 min. Subsequently, the sediments were collected and re-dispersed in ultrapure water (0.65 mL). The purification processes were then repeated to completely remove the surfactant residue. After purification, a portion of the MoS$_2$ sediments from 0.65 mL dispersions was freeze-dried and weighed to calculate the yield of MoS$_2$ nanosheets (1.8%) based on the subtraction method. Finally, the purified sediments

were re-dispersed in ultrapure water to prepare a stock MoS$_2$ dispersion (200 μg mL$^{-1}$).

### Synthesis of Au/MoS$_2$ hybrid nanocrystals

A seeding method was used for the growth of the gold nanocrystals on exfoliated MoS$_2$ layers. First, growth solutions with gold ions were prepared. The CTAB (100 mM) stock solution was heated at 30 °C to prevent crystallization. At room temperature, 8.00 mL of CTAB (100 mM) was added to a sample bottle, and the whole system was continuously stirred at 500 rpm using a magnetic stirrer. Next, 800 μL of HAuCl$_4$ (25 mM) was added to form a turbid solution of the [AuBr$_4$]$^-$ complex. Noticeably, the color of the mixture subsequently changed from faint yellow to pale brown. Ultrapure water (39.50 mL) was injected immediately, and the solution adopted a clear golden color. Au$^{3+}$ was then rapidly reduced to Au$^+$ by rapid injection of 4.75 mL of 100 mM fresh AA solution. The growth of gold nanostructures began immediately once the seeds of MoS$_2$ layers were added. Cracked polyhedra were prepared by adding 1200 μL of the exfoliated MoS$_2$ seed solution into the reaction system. The growth of chiral dendritic nanocrystals was initiated by the addition of 1200 μL of seed solution containing amino acids (cysteine) and exfoliated MoS$_2$ into the growth solution. Furthermore, the reaction system was incubated in a water bath at 40 °C for 30 min. The green MoS$_2$ solution gradually turned to bluish violet and then blue with a remarkable scattering effect. Finally, the solution was centrifuged twice (10965 × $g$ for 10 min) to remove unreacted reagents, and the sediments were redispersed in the CTAB solution (1 mM) for subsequent experiments.

### Arresting growth via ligand exchange and solvent transfer

To identify the reaction processes and intermediate morphologies of the growing nanostructures, their growth was arrested at different times via ligand exchange and solvent transfer[62,63]. When the reaction was conducted for different times (20 to 480 s), a 2.0 mL portion of the reaction solution was pipetted and rapidly added into a tetrahydrofuran solution (2.0 mL) containing thiolated polystyrene (4 × 10$^{-4}$ M). Subsequently, the mixture was shaken vigorously for 1 min and became turbid. After centrifugation (3948 × $g$, 5 min), the sediment was collected and completely redispersed in toluene via sonication. To characterize samples, UV-vis-NIR optical absorption spectra of nanostructures dispersed into toluene were recorded. Additionally, the morphologies of the nanostructures were captured by TEM. A total of 2.0 μl of nanostructure/toluene solution, which was freshly prepared, was dropped onto a 200-mesh carbon-coated Cu grid and dried by blowing with N$_2$ gas for 30 min. To avoid morphological changes and biased analyses, optical absorption and TEM data were obtained immediately after dispersing the nanostructures in toluene.

### Preparation of doped hydrogels and drug-loaded hydrogels

The hydrogel in the sample bottle was presented. The sample bottle was preincubated in a water bath (60 °C). First, a 1.0 mL portion of heated (90 °C) agarose stock solution and a diluted dispersion of nanostructures (0.9 mL) were added into the sample bottle. Subsequently, ofloxacin was dissolved into a 1.0 M HCl solution, and a 0.1 mL portion of the drug solution was added into the mixture of agarose and the nanostructures. The whole operation was conducted at 60 °C to solidify the agarose. The additives were mixed and then left to cool to 25 °C, and they formed the samples for the drug-loaded hydrogels. In the control sample, the ofloxacin solution was replaced by a pure HCl solution (1.0 M), and the other operations remained constant. To prepare hydrogel tablets for the antimicrobial experiments, a 200 μL portion of the dispersed mixture was subsequently injected into the matrix assay. The matrix assay consisted of a series of holes with a diameter of 1 cm. After cooling to 25 °C, the mixed solution coagulated and formed hydrogels. The hydrogel tablets were collected via pushing with a smooth-ended glass rod. Using the concentrations of drugs

and nanomaterials in the stock solution, hydrogel samples were prepared with different amounts of drugs and nanomaterials loaded.

## Antimicrobial experiments

Hydrogel flakes with the diameters of 1.0 cm were prepared for the assays. The agarose solution (2.5%) was heated by a microwave oven. To prevent the cooling of the mixture, all reagents were preincubated at 50 °C, and the operations described below were also conducted in a water bath (50 °C). First, the levofloxacin powders were dispersed into an HCl solution (1.0 M). A 0.4 mL portion of the nanostructures solution was dispersed into ultrapure water (0.4 mL). The heated agarose solution (2.5%, 1.0 mL) was added to the nanomaterial dispersion and totally mixed using a pipette. The levofloxacin solution (0.2 mL) was finally added to the agarose and nanomaterials mixture. A 200 μL portion of the warm mixture (50 °C) was transferred into the matrix, which was cooled for preparation of the hydrogels. Two strategies were utilized for assessing the antimicrobial efficiency. First, a coincubation route was developed. The nanomaterials were preincubated with *S. aureus* (1 × 10⁶ CFU/mL) for 12 h (37 °C). After incubation, the bacterial samples were spin-coated on the agarose substrate with the culture medium. Subsequently, the treated samples were incubated at 37 °C for 20 h. More importantly, antimicrobial experiments on nanomaterial hydrogels with diameters of 1.0 cm, were performed, and the antimicrobial efficiencies were determined from the sizes of the bacteriostasis circles.

## Sustained antimicrobial studies

After a tablet was prepared, the antibacterial activity of the tablet was determined by the size of the inhibition zones formed using *S. aureus*, *E. coli*, and *P. aeruginosa*. The bacterial suspension with a concentration of approximately 5 × 10⁷ CFU/mL was inoculated onto LB nutrient agarose culture plates. After that, the tablets are placed on the plates. After 24 h of incubation, a photograph was recorded (1st Period), and the tablets were carefully transplanted onto a new LB nutrient agarose culture plate and cultured for the next 24 h (2nd Period). Subsequently, to assess the sustained performance, the antimicrobial experiments were repeated for two other periods of 24 h (3rd Period and 4th Period). The sizes of the inhibition zones around the tablets were recorded.

## IR-activating antimicrobial performance

To study photothermally-induced drug release, hydrogel tablets loaded with different concentrations of levofloxacin were placed in LB nutrient agarose culture plates and subsequently cultured for 1 h. After that, the hydrogel tablets were treated with IR irradiation (1.0 min). As control samples, hydrogel tablets without IR treatment were studied. Subsequently, the bacterial culture systems were further incubated for 23 h. The bacteriostasis circles were recorded for assessment. The efficiencies of IR-activated tablets were assessed for sustained antimicrobial performance. In terms of 1st Period, 2nd Period, 3rd Period, 4th Period, IR treatments were only applied (1.0 min) in the last 24 h period, and the treatments of the former period/periods remained constant.

## Computational details

All calculations were carried out with the Vienna ab initio simulation package (VASP) version 5.4.4[64]. A plane-wave basis set with a cut-off energy of 400 eV was used to represent the wave functions of valence electrons, and the projector-augmented wave (PAW) method was adopted for the core electrons[65]. The convergence criterion for the ion self-consistent iteration was set as −2 × 10⁻² eV[66]. In this study, we used the Perdew-Becke-Erzenhof (PBE) exchange-correlation functional. The $\gamma$ k-point sampling grid 2 × 1 × 1 was adopted for geometric optimization, and a 2 × 2 × 2 grid was used for DOS calculations. To expose the crystal faces, we cut the crystal faces of Au separately along the

{321}ᴿ and {321}ˢ planes. All calculations were performed with 8 layers of 2 × 1 periodic cells (atoms with the bottom four layers fixed). The adsorption energies for adsorbates were calculated as:

$$E_{ad} = E_{substrate-adsorbate} - E_{substrate} - E_{adsorbate} \qquad (1)$$

## Statistical analysis

Statistical data are presented as mean values ± SD. Statistical analysis was performed using Origin 2018 software. All statistical data were collected from experiments with three replicates ($n = 3$). The value of $P < 0.05$ was considered significant.

## Reporting summary

Further information on research design is available in the Nature Portfolio Reporting Summary linked to this article.

## Data availability

All data supporting the findings of this study are available within the article, source data, and its Supplementary Information. Any other data are available from the authors upon request. Source data are provided with this paper.

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

## Acknowledgements

B.L.L. acknowledges the funding provided by the National Natural Science Foundation of China (No. 22004103), Innovation Research 2035 Pilot Plan of Southwest University (SWU-XDZD22011), Fundamental Research Funds for the Central Universities (SWU-KT22028), and the Municipal Science Foundation of Chongqing City (No. cstc2021jcyj-msxmX0295). J.J.L. acknowledges the National Training Program of Innovation and Entrepreneurship for Undergraduates (202210635087). The authors appreciate Prof. Huawei He (Biological Science Research Center, Southwest University, China) for the CD spectra measurements, Prof. Wei Shen and Na Zhang (School of Chemistry and Chemical Engineering, Southwest University, China) for the theory simulation and TEM tests, respectively, and Dr. Wenjuan Yuan (Center for Electron Microscopy, Tianjin University of Technology) for the TEM Tomography.

## Author contributions

B.L.L., D.T.L., and N.B.L. conceived the project and designed experiments. B.L.L., J.J.L., H.L.Z., and H.Q. performed the experimental studies. B.L.L., Q.M.Z., and L.B.Z. conducted and analyzed the theoretical computation. B.L.L. and N.B.L. analyzed the experimental results. B.L.L., H.Q.L., D.T.L., and N.B.L. co-wrote the main parts of the paper. All authors discussed and edited the manuscript and gave approval to the final version of the manuscript.

## Competing interests

The authors declare no competing interests.
