## [Peer Review File · Nature Communications]

Reviewer comments, initial review

Reviewer #1 (Remarks to the Author):

The manuscript titled "Dendritic nanocrystals inheriting the chirality from amino acids at heterostructural interfaces" described that amino acids with chiral conformations modulate the asymmetric growth of gold nanoarchitectures at the MoS₂ heterostructural interfaces, and the cysteine-endowed heterostructural architectures encode chirality into nanocrystals. It is an interesting topic, but the novelty of this article isn't enough. Meanwhile, this manuscript isn't well organized. I strongly suggest that the authors carefully check and correct the manuscript and clarify what the abbreviations mean. It is hard for the reviewer and reader to understand. The questions are as follows:

1. Why the sizes in Figure 1b-d are different? The size of nanostructures in Figure 1bc looks much bigger than the nanostructures in figure 1d.
2. The authors should clearly explain the synthesis and characterization of hybrid nanocrystals in the main manuscript. The supplementary figures were not well explained in the main manuscript, so it is hard to understand the key points and catch up with the key conclusion.
3. What is V1 in line 165?
4. The authors should add the spectra of (R)-DNCs in figure 3b.
5. What is (D)-DNCs in line 286?
6. What are (S)-HNCs and (R)-HNCs in figure 4f?
7. Statistical analysis also should be conducted to show the significance of difference for the release ratio in figure 4i, figure 5d, and f.
8. The authors claimed in line 307 that 'Both (S) and (R)-enantiomers of dendritic nanostructures exhibited comparable antimicrobial activities.' Why do nanostructures have antimicrobial activity? In this case, the levofloxacin-free agarose tablets with the doping of (R)-DNCs and (S)-DNCs should be added as a control group in all antimicrobial experiments.
9. Whether the photothermal properties contribute to the killing of bacteria? Any data to explain?
10. the authors claimed that 'DNCs exhibited weak antimicrobial efficiency.' However, in supplementary Fig. 10, the authors claimed that 'both the (S)-enantiomer and (R)-enantiomer of dendritic nanostructures exhibited the attractive antimicrobial performance.' I am confused.
11. *S. aureus* has been tested to show the antimicrobial effect. How about another type of bacteria? More bacteria types should be involved to prove the antimicrobial effect.

Reviewer #2 (Remarks to the Author):

In this manuscript, the authors demonstrated chiral Au-MoS₂ heterostructures with aqueous-based growth method using L- and D- cysteine molecules. Experimental and theoretical results showed that the dendritic nanocrystals induced asymmetric photothermal properties. In addition, the authors demonstrated sustained drug release of dendritic Au-MoS₂ heterostructures incorporated in hydrogel matrix when irradiated with NIR laser. However, the authors provide insufficient explanations for the enantioselectivity of photothermal phenomena and drug release and their origins. In addition, we think that additional experiments and clear demonstration should be conducted to support authors' claim. Therefore, we would like to recommend the authors consider the items below to make the manuscript stronger to fulfill the high standard of Nature Communications.

- 1) In Figure 1, at 0.2 μ M and 2 μ M cysteine concentration, the CD response when using L-cysteine was greater than when using D-cysteine. However, in 1.2 μ M and 1.4 μ M, relatively similar main peak intensities were observed. Is there an optimum range of amino acid concentration for mirror symmetric CD response?
- 2) In addition, in CD spectra in Figure 1, negative CD dips were observed near 400 nm for all chiral crystals directed by L- and D-cysteine at all cysteine concentrations. It seems the authors need to explain the origin of this CD peak.
- 3) How heterostructures grow from MoS₂ seeds with mixed sizes. Do dendritic crystals grow by aggregation of several MoS₂ seeded Au during the growth process?

- 4) In this manuscript, the authors reported that dendritic branched structures are generated only in amino acids and small molecules containing thiol group. Does the absence of branched structures in gsh and BSA containing thiol groups suggest that the development of dendritic structures is related to molecular size?
- 5) In the photothermal experiment of the dendritic nanocrystal in Figure 3c, what is the reason of the larger temperature difference between R- and S-dendritic nanocrystals when introduced into hydrogel than aerogel?
- 6) To compare the chirality of synthesized nanoparticles, it is necessary to calculate the g-factor. Assuming that the CD spectra in Figure 1 measures nanoparticles of the same concentration, it seems that the CD signal in the visible region increases as the concentration of cysteine included in the synthesis increases. How does injection of cysteine over 2 μm affect particle chirality?
- 7) The authors need to provide the difference between R- and S-dendritic nanocrystals in the demonstrated photothermal effect and drug release efficiency.

Reviewer #3 (Remarks to the Author):

This manuscript deals with the synthesis of chiral Au nanostructures using MoS₂ as templates and amino acids as their application for photothermal-response drug release. It is an interesting and relevant work, but the manuscript is very confused and difficult to follow. Besides, part of the discussion is not very well supported by experimental data I think that the authors must perform a deep revision of the manuscript. Therefore my consideration is "not publish".

Comments:

- 1) Regarding the synthesis of the Au nanostructures in the presence of MoS₂ and in the presence or absence of amino acids. The discussion is based on the changes in the color and it is not very scientific. Based on the dependence of the optical properties in the size and shape of the Au nanoparticles, the authors to support their assumptions must perform the analysis of their optical properties using UV-Vis spectroscopy. Please, also revise the text, line 83, Figure 1a does not match there. Besides, the authors must include SEM/TEM analysis of the particles obtained in the absence of MoS₂.
- 2) The authors mention the importance of the size of MoS₂, nevertheless no information about that is included.
- 3) In line 87, the sentence "It was supposed that discriminated growths of nanostructures, induced by the small molecules of amino acids in the homogeneous solution, were activated" It is not supported by any data at this point.
- 4) In line 91, it is written that the particles are like a paper crane, from my point of view there is no need of including that. The particles are multibranched as showed the SEM and TEM analysis.
- 5) The CD is closely related to the absorption properties of the nanoparticles therefore for me the optical properties should be analyzed together and not afterward. Anyway, I do not agree with the analysis of the optical properties showed in lines 119-121, it is very poor. The same when analyzing the amount of L-cys. Besides, how is the g value obtained for these nanostructures? Besides, no discussion is performed about the effect of the amount of Cys.
- 6) The discussion about the changes in the morphology of the nanostructures is supported by TEM and SEM images. It is correct but the conclusions are overestimated. For me, It is not clear the slight zigzag that the authors observed on the branch of BNCs. Even I am not able to see the differences between L-Cys and D-Cys (2j and 2k). To claim that the authors should perform 3D tomography.
- 7) Regarding the use of other compounds such as BSA or GSH, it is not clear to me which is the aim of this work obtained branched nanoparticles or chiral nanoparticles with a high chiral

response. Since no discussion about the chirality is included in the main manuscript. Moreover, for those experiments why was a concentration of 1.6 microM used?

8) Regarding the mechanism, I suggest improving the discussion and performing a time-resolved TEM analysis to observe the evolution of the nanostructures during its formation.

9) Regarding the discriminated growth, the structure of the resulting nanoparticle has not been analyzed in detail. Therefore I do not understand why authors in line 174 claim that the particles are stellated octahedron differentiated by [321] facets. It is not supported by the experimental data. As mention before, it is difficult to believe for the data shown that BNCs have L-cys and D-cys structures (fig J and K).

10) Regarding the experiments of the nanostructures for photothermal-response drug release, I do not know the meaning of the sentence of Line 225-227 "The differential conformation of BNCs and DNCs exhibit the consistent absorption performances (supplementary Fig 7)". Besides, why they selected the DNCs nanoparticles for the photothermal experiments (figure 3c) and not the other chiral nanoparticles obtained.

11) In figure 4f it is showed the loading efficiency of the hydrogels, right? Then I do not know why in the discussion it is explained as release ratios (lines 268-270). The discussion about the differences in the release among the different nanoparticles is very confusing and the message is not clear.

Reviewer #4 (Remarks to the Author):

The manuscript "Dendritic nanocrystals inheriting the chirality from amino acids at heterostructural interfaces" by the anonymous authors is on the very topical subject of chirality transfer.

The introduction lacks structure. The general background is not presented clearly. There is no clearly defined problem and the question(s) the authors are trying to answer is not presented. The key concepts are not introduced (explain: what is chirality? What is chirality transfer? What is quantum confinement-determined optical activity? Why are amino acids important/essential? When talking about "chiral recognition capacity", who is recognizing what and why should the reader know about it? What are metamaterials?) The state of the current literature is not adequately surveyed – this is a paper about chirality transfer, there should be many references on chirality transfer with clear explanations: e.g. "such and such demonstrated chirality transfer from this to that". Instead, there is a collection of vaguely related sentences and references, generally on the topic of chirality and some references on chirality transfer but without clear explanations or a unifying narrative.

The manuscript it not written in clear English, e.g. "Nevertheless, their potential performances are unfortunately inactivated."

The interaction between cysteine and MoS₂ is intriguing and potentially interesting. The way the data are presented though is unclear. What is this origami image? Why is it there? All acronyms should be defined. In figure 1, are the CD spectra reproducible or have they been hand-picked to show opposite CD? Are the CD spectra from single nanoparticles or from the whole sample?

Figure 2 is very interesting and could almost be a paper by itself, demonstrating that these are separate stages indeed, and what the conditions for each case are.

Figure 3 is very unclear and the caption does not help much to understand what is what. Again, if sufficiently explained and backed up by scientific evidence at every statement, this might be a paper by itself. Where is the cysteine?

Figure 4 could likely also be a paper by itself, with more data to actually show that this is happening as the authors say it does. The same comment for Figure 5.

I assume that the authors have actually read reference 2 (and all the references they cite). Does this paper really illustrate chirality transfer or does it say that there is no chirality transfer in the Te systems?

Overall, this is an intriguing work and it might well be interesting to a broad community of readers. Unfortunately, the presentation is poor to the point of incomprehensible. The lack of clarity in English results in a lack of rigor, so some of the scientific claims seem unsubstantiated. I also suspect that the authors are trying to put too much work into a single paper, without satisfying the requirements for sufficient detail. The authors might be well advised to split this work in 3 to 5 separate but related manuscripts, making sure that every single scientific claim is supported by direct evidence.

Point-to-Point Responses to Reviewers' Comments (NCOMMS-22-07182)

We would like to express our gratitude to reviewers for your encouraging and constructive comments. We strongly believe that this manuscript will not be in a better form without those comments and suggestions. All changes in the revised manuscript are marked with red font for tracking purposes. Point-to-point responses to comments are shown below:

Reviewer #1: *The manuscript titled “Dendritic nanocrystals inheriting the chirality from amino acids at heterostructural interfaces” described that amino acids with chiral conformations modulate the asymmetric growth of gold nanoarchitectures at the MoS₂ heterostructural interfaces, and the cysteine-endowed heterostructural architectures encode chirality into nanocrystals. It is an interesting topic, but the novelty of this article isn't enough. Meanwhile, this manuscript isn't well organized. I strongly suggest that the authors carefully check and correct the manuscript and clarify what the abbreviations mean. It is hard for the reviewer and reader to understand. The questions are as follows:*

Thanks for your constructive comments. We have re-organized our manuscript, and in order to reach the high standard of *Nat. Commun.*, we also highlighted our novelty, showing the chiral transfer of amino acids to plasmonic gold nanostructures at heterostructural MoS₂ surfaces. Additionally, the chiral recognition of natural molecules was also inherited to highly faceted Au/MoS₂ nanoarchitectures, performing discriminated interaction role to enantiomer drugs. The abbreviation words were carefully checked and re-clarified as required. Overall, the point-to-point revision list and responses are shown below:

Q1. *Why the sizes in Figure 1b-d are different? The size of nanostructures in Figure 1bc looks much bigger than the nanostructures in figure 1d.*

Response: The MoS₂ layer seeds were prepared from a top-to-down liquid-phase exfoliation route, and nanosheets with size diversity were collected, which might result in the size differences of those as-synthesized Au/MoS₂ nano-heterostructures. Consequently, the sizes of Au/MoS₂ nanostructures in one sample could change from

90 to 130 nm. We seriously attach importance to this query of Reviewer #1. After the check with our initial SEM and TEM data, we correct the mistake of the TEM scale bar shown in Fig. 2f. In addition, the scale bars of TEM and SEM figures were also united (50 nm). According to the comment, the statistic results indicated that the maximum diameter of the branched nanomaterials ranging from 90 nm to 130 nm, and the corresponding size information was added in lines 132-135 (page 5) in the revised manuscript.

Q2. The authors should clearly explain the synthesis and characterization of hybrid nanocrystals in the main manuscript. The supplementary figures were not well explained in the main manuscript, so it is hard to understand the key points and catch up with the key conclusion.

Response: Appreciate reviewer #1 for this suggestion. We have transferred these key points about the synthesis and characterization of as-synthesized hybrid nanocrystals from the Supplementary Information to the main content. More description about the supplementary figures has been added in the revised manuscript. The supplementary content about the synthesis reaction of nano-structures can be found in lines 109-123 (pages 4 and 5) in the revised manuscript, which was labeled with red color.

Q3. What is V1 in line 165?

Response: In this manuscript, the *V1* represents to the rate of Au atom deposition on MoS₂ edges. According to the comment, the description about the atomic deposition and diffusion on MoS₂ layers was revised for a better understanding, which can be found in lines 226 and 227 (page 9) in the revised manuscript.

Q4. The authors should add the spectra of (R)-DNCs in figure 3b.

Response: According to the comments, the vis-NIR absorption spectrum of (R)-DNCs-2 was added in the revised manuscript (Fig. 5b). In order to present the better understanding, the Vis-NIR absorption spectra of these slightly branched nanocrystals, referring to (S)-DNCs-1 and (R)-DNCs-1, were also added in the revised manuscript. Relevant description has been shown in lines 316-319 (page 13)

in the revised manuscript.

Q5. What is (D)-DNCs in line 286?

Response: Thanks for your check. We have changed the (D)-DNCs to (R)-DNCs in the line 363 (page 15) in the revised manuscript. Sorry for our typos.

Q6. What are (S)-HNCs and (R)-HNCs in figure 4f?

Response: In the previous version of our manuscript, the HNCs were wrongly labeled, which must be the branched nanocrystals (BNCs). The BNCs were synthesized in the presence of exfoliated MoS₂ and 1.2 μM cysteine. In our revised manuscript, we changed the definition of BNCs to the DNCs-1. Of significance, due to the decreases of cysteine concentrations, the branched degree of DNCs-1 was lower than DNCs-2. Moreover, the chiral DNCs-1 nanostructures were identified and labeled as (S)-DNCs-1 and (R)-DNCs-1 as the uses of L-cysteine and D-cysteine reagents, respectively. Sorry for our typos. The figure (Fig. 5c) had been revised, and more information about the description of chiral DNCs-1 and DNCs-2 was added in lines 316-331 (page 13) in the revised manuscript.

Q7. Statistical analysis also should be conducted to show the significance of difference for the release ratio in figure 4i, figure 5d, and f.

Response: Thanks for your comment. We revised the figure number in the revised manuscript. Statistical analysis was conducted and the information is shown in figures and figure captions (Fig. 5i, Fig. 6f, h, j, k) of the revised manuscript.

Q8. The authors claimed in line 307 that “Both (S) and (R)-enantiomers of dendritic nanostructures exhibited comparable antimicrobial activities.” Why do nanostructures have antimicrobial activity? In this case, the levofloxacin-free agarose tablets with the doping of (R)-DNCs and (S)-DNCs should be added as a control group in all antimicrobial experiments.

Response: From the co-incubation and bacteriostasis circle antimicrobial experiments, we found that both (R)-DNCs-2 and (S)-DNCs-2 exhibited comparable antimicrobial activities when the concentrations of nanostructures were over than 150

$\mu\text{g mL}^{-1}$. Previous works about the gold nanostructures indicated that the antimicrobial performance could be attributed to reactive oxygen species (ROS)-induced destroy of bacterial cell membrane.^{1,2} It was highly worthy to conduct the further studies of antimicrobial impacts using our developed nanostructures. Thanks for your suggestions. Meanwhile, the changes in diameters of bacteriostasis circle were listed, and we assessed the antimicrobial performance of hydrogel samples in comparison to the levofloxacin-free agarose tablets. Thanks for your reminding and constructive suggestions. Relevant description has been added in lines 390-394 (page 16) and lines 399-408 (page 16) in the revised manuscript.

Q9. Whether the photothermal properties contribute to the killing of bacteria? Any data to explain?

Response: Thanks for your question. Definitely, we should conduct the assessment of photothermal efficiency based on the levofloxacin-free agarose tablets with the doping of (*R*)-DNCs-2 and (*S*)-DNCs-2. It was found that the nanomaterials were loaded in the hydrogel, rather than the homogenous solution, the photothermal antimicrobial performance was low. Once the tablets were loaded with levofloxacin, the DNCs hydrogels exhibited highly enhanced antimicrobial impacts to *S. aureus*. Significantly, the responsively improved antimicrobial performances could be attributed to the photothermal-induced molecular release, rather than the photothermal killing (Fig. R1). In order to support the photothermal activation in drug release, the Fig. R1 was added in the revised manuscript (Fig. 6j), and relevant description has been added in lines 427-431 (page 17) in the revised manuscript.

[1] Zheng, K., Setyawati, M. I., Leong, D. T. & Xie, J. P. Surface ligand chemistry of gold nanoclusters determines their antimicrobial ability. *Chem. Mater.* **30**, 2800-2808 (2018).

[2] Sarker S. R. *et al.* Functionalization of elongated tetrahedral Au nanoparticles and their antimicrobial activity assay. *ACS Appl. Mater. Interfaces* **11**, 13450-13459 (2019).

Fig. R1 Changes of BC values of AGH, (S)-DNCs-2/AGH, and (R)-DNCs-2/AGH tablets before and after the IR activation. The tablets are loaded with differential concentrations of levofloxacin. Data are means \pm SD, n = 3, Student's t-test, p < 0.1, *Significant against the hydrogel samples without the levofloxacin loading, #Significant against the AGH samples.

Q10. The authors claimed that “DNCs exhibited weak antimicrobial efficiency.” However, in supplementary Fig. 10, the authors claimed that ‘both the (S)-enantiomer and (R)-enantiomer of dendritic nanostructures exhibited the attractive antimicrobial performance.’ I am confused.

Response: Sorry for our inappropriate description in the original manuscript. We conducted the antimicrobial researches based on the co-incubation method, and the results are shown in the supplementary information. (S)-DNCs-2 and (R)-DNCs-2 were found to present comparable antimicrobial activities in the co-incubation antimicrobial experiments, but the nanostructure-doped hydrogel tablets exhibit relatively weak bacteriostasis circle antimicrobial performance when they are compared with the levofloxacin-loaded hydrogel samples. Even the antimicrobial performances of as-synthesized Au/MoS₂ nano-heterostructures were limited, considering the side effects and drug resistance, the nanostructures without the involvement of organic drugs were highly encouraged to go further exploitation.³ In

[3] Zheng, K. Y., Setyawati, M. I., Leong, D. T. & Xie, J. P. Antimicrobial gold nanoclusters.

order to avoid misconceptions, we have revised relevant description, shown in lines 390-394 and lines 399-405 (page 16), in the revised manuscript.

Q11. *S. aureus* has been tested to show the antimicrobial effect. How about another type of bacteria? More bacteria types should be involved to prove the antimicrobial effect.

Response: According to the comment, the anti-bacterial performance of our developed sustainable and IR-responsive to *E. coli*, and *P. aeruginosa* was supplemented. First, in the presence of levofloxacin, Au/MoS₂ DNCs-2 nanostructures exhibited differential impacts to the bacterial killing, which were varied as the differences of bacterial species. Among those three kinds of bacteria, DNCs-2 exhibited the best antimicrobial performance to *S. aureus*, but performed the weakest impact to *P. aeruginosa* (Fig. R2a, b, c). With the loading of levofloxacin, the hybrid hydrogels showed enhanced efficiencies to these three kinds of bacterial. Additionally, the sustainable levofloxacin release systems worked well, indicating that the levofloxacin-loaded nanostructure hydrogels could promote to the multi-bacterial killing systems (Fig. R3). The relevant description has been added in lines 399-408 (page 16), lines 412-420 (page 17) in the revised manuscript. Fig. R2 was presented as Fig. 6c-e in the revised manuscript, and Fig. R3 was added in the supplementary information (Supplementary Fig. 10).

Fig. R2 a-c Pictures indicating the *S. aureus* (a), *E. coli* (b), *P. aeruginosa* (c) bacteriostatic circles of diverse tablets which consisted of diverse concentrations of levofloxacin (unit: $\mu\text{g mL}^{-1}$).

Fig. R3 a, b Pictures showing the antimicrobial performance of AGH, (R)-DNCs-2 AGH, and (S)-DNCs-2 AGH to *E. coli* and *P. aeruginosa* at period of D1 and D4 in the sustainable drug release experiment. **c, d** Statistic results indicating the sustainable release results through the analysis of day-dependent bacteriostatic circles of *E. coli* (**c**) and *P. aeruginosa* (**d**) in terms of diverse levofloxacin-loaded hydrogel tablets. The curves in **c, d** exhibit the bacteriostatic circles of hydrogel tablets without levofloxacin.

Reviewer #2: In this manuscript, the authors demonstrated chiral Au-MoS₂ heterostructures with aqueous-based growth method using L- and D- cysteine molecules. Experimental and theoretical results showed that the dendritic nanocrystals induced asymmetric photothermal properties. In addition, the authors demonstrated sustained drug release of dendritic Au-MoS₂ heterostructures incorporated in hydrogel matrix when irradiated with NIR laser. However, the authors provide insufficient explanations for the enantioselectivity of photothermal phenomena and drug release and their origins. In addition, we think that additional experiments and clear demonstration should be conducted to support authors' claim. Therefore, we would like to recommend the authors consider the items below to make the manuscript stronger to fulfill the high standard of Nature Communications.

Response: Thank you very much for your valuable points and precious effort on improving this manuscript. Supplementary works about the studies of enantioselectivity of photothermal phenomenon and drug release had been conducted and relevant description was added in the revised manuscript. The origins of enantioselectivity were proposed, and extra experiments were designed to identify our claim. Thanks for your kind and constructive suggestions. Point-to-point responses are listed below:

Q1. In Figure 1, at 0.2 μ M and 2 μ M cysteine concentration, the CD response when using L-cysteine was greater than when using D-cysteine. However, in 1.2 μ M and 1.4 μ M, relatively similar main peak intensities were observed. Is there an optimum range of amino acid concentration for mirror symmetric CD response?

Response: We found that when the concentrations of cysteine were fixed, the CD responses of nanostructures were consistent when different enantiomers of cysteine were used. We conducted the supplementary experiments for the optimization of cysteine concentrations. Compared to the control sample in the absence of cysteine, when the concentration is lower than 0.2 μ M, almost none of any morphological changes could be found on Au/MoS₂ nanostructures. In addition, we also exploited the reactions when the cysteine concentrations were higher than 2.0 μ M. The morphologies and optical absorption of as-formed nanostructures kept unchanged,

but they also exhibited the mirror symmetric CD responses (Fig. 1f in the revised manuscript). In our previous work, we found that high-concentration cysteine could also provide reduction impacts to Au⁺ ions,⁴ and thus the growth mechanism of Au/MoS₂ nanostructures in the presence of millimolar cysteine might be complex. It is worthy to go further exploration about high-concentration cysteine in our further works.

Q2. In addition, in CD spectra in Figure 1, negative CD dips were observed near 400 nm for all chiral crystals directed by L- and D-cysteine at all cysteine concentrations. It seems the authors need to explain the origin of this CD peak.

Response: In our previous experimental results, the diluted solutions of nanoparticle dispersion were used for CD measurements. In order to give responses to reviewer's concerns, we re-measured all CD spectra (Fig. 1f in the revised manuscript) using the reaction solutions directly, and it was found that the CD responses remarkably increased. More importantly, the negative CD dips disappeared in terms of all samples. Thus, the CD dips at 400 nm could be attributed to the background noises. The origin of CD peaks could be attributed to the surface plasmonic resonance (SPR) absorption of chiral Au/MoS₂ nanoparticles to the right/left circularly polarized light (CPL).⁵ Of significance, chiral molecules can induce a plasmonic circular dichroism (PCD) response in the SPR region, and strong PCD signal can be generated via the coupling of plasmonic dipoles in chiral metal nanostructures.^{6, 7} According to the comments, we have added the origin of the CD peaks in the revised manuscript,

[4] Du, X. J. *et al.* Plasmonic gold nanoparticles stain hydrogels for the portable and high-throughput monitoring of mercury ions. *Environ. Sci. Technol.* **56**, 1041-1052 (2022).

[5] Lee, H.-E. *et al.* Cysteine-encoded chirality evolution in plasmonic rhombic dodecahedral gold nanoparticles. *Nat. Commun.* **11**, 263 (2020).

[6] Zhang, G. C. *et al.* Tuning the morphology and chiroptical properties of discrete gold nanorods with amino acids. *Angew. Chem. Int. Ed.* **57**, 16452-16457 (2018).

[7] Lan, X. *et al.* Au nanorod helical superstructures with designed chirality. *J. Am. Chem. Soc.* **137**, 457-462 (2015).

shown in lines 142-155 (pages 5 and 6) in the revised manuscript.

Q3. How heterostructures grow from MoS₂ seeds with mixed sizes. Do dendritic crystals grow by aggregation of several MoS₂ seeded Au during the growth process?

Response: The time-resolved TEM for identification of intermediate structures was characterized. Herein, the TEM characterizations were based on the separation of nanoarchitectures at different reaction times. From the transition state of dendritic nanocrystals, it was found that the nanomaterials were synthesized from one nanosheets. The size diversity of heterostructures could be attributed to the differential sizes of MoS₂ seeds which were obtained from the top-to-down exfoliation methods. From the composition of transition samples, it was found the gold nanostructure was grown on one thin nanosheets (Fig. R4), indicating that the growth of plasmonics was supported by the individual MoS₂ seeds, rather than the aggregation of Au/MoS₂ nano-heterostructures. According to the comments, we have added the relevant description in lines 207-223 (page 9) in the revised manuscript, highlighting the role of MoS₂ in directing the growth of Au and MoS₂ nano-heterostructures. Fig. R4 in the response letter was added in the revised manuscript (Fig. 3e, f).

Fig. R4 **a** and **b** TEM, HAADF, and EDX element mapping scan of Au/MoS₂ nanostructures which are collected after the reaction for 50 s in the presence of 1.2 μ M L-Cys. The scale bars in **a** and **b** are 50 nm.

Q4. In this manuscript, the authors reported that dendritic branched structures are generated only in amino acids and small molecules containing thiol group. Does the absence of branched structures in GSH and BSA containing thiol groups suggest that the development of dendritic structures is related to molecular size?

Response: Initially, from our TEM experiment data, it was found that the increasing molecular sizes contributed to the decreases of branched degrees to the Au/MoS₂ heterostructures. Nevertheless, none of strong evidence could be provided to identify the influence of molecule sizes to the differences of branched structures. In order to avoid confuse to readers about our novelty and key points, we revised the description and reports about the GSH and BSA, highlighting the function of thiol groups to the anisotropic growth of Au nanoarchitectures. Instead, the experimental results about other chiral amino acids with and without thiol groups had been added, focusing on the studies of differential interaction of amino acids to the branched structures. Relevant description had been added in lines 233-242 (pages 9 and 10) in the revised manuscript. Regarding to this comment, we will go further about the studies of molecular size to the influence of heterostructure growth. Appreciate for the suggestions from Reviewer 2# and Reviewer 4#.

Q5. *In the photothermal experiment of the dendritic nanocrystal in Figure 3c, what is the reason of the larger temperature difference between R- and S-dendritic nanocrystals when introduced into hydrogel than aerogel?*

Response: From our current experiment data, it was found that the (S)-DNCs hydrogel owned slightly higher temperature than (R)-DNCs hydrogel. Nevertheless, the changes were lower than 5%, which might be caused by the unexpected deviation. We could not get the conclusion that the R- and S- DNCs perform any differences in photothermal efficiency when those materials were loaded in the both hydrogel and aerogel. In order to avoid misconceptions, the experimental results were highlighted in lines 323-328 (page 13) in the revised manuscript. The heat is trapped in the aerogel as air in the pores is a poorer heat conductor than water in the hydrogel's pores, thus increasing the temperature of the aerogel. According to the comment, the description about the high photothermal efficiency of hydrogel was added in lines 319-322 (page 13) in the revised manuscript.

Q6. *To compare the chirality of synthesized nanoparticles, it is necessary to calculate the g-factor. Assuming that the CD spectra in Figure 1 measures nanoparticles of the*

same concentration, it seems that the CD signal in the visible region increases as the concentration of cysteine included in the synthesis increases. How does injection of cysteine over 2 μM affect particle chirality?

Response: Considering the differential molecular structures of amino acid enantiomers, L-Cys is replaced by D-cysteine (D-Cys) and a series of Au/MoS₂ products are collected with the increasing concentrations of additive L/D-Cys. The chirality performance of branched nanostructures dependent on the enantiomer classification of cysteine was studied by circular dichroism (CD) spectra (Fig. R5). The products exhibit two pairs of symmetric CD peaks (how to explain the different directions). The higher CD peaks shift from 550 nm to near 900 nm with the concentrations of L-Cys and D-Cys increasing from 0.4 to 2.0 μM , exhibiting the consistent regulation of optical absorption performance induced by the localized surface plasmon resonance (LSPR) characteristics of nano-plasmonics. Remarkably, in comparison to previous chiral plasmonic works, the large shifts of CD peaks are confirmed, which could be ascribed to remarkable morphological diversity of symmetric nano-plasmonics based on the synergistic effects of exfoliated MoS₂ and cysteine enantiomers. The *g*-factor values were analyzed all over the studied concentrations of cysteine (Fig. R6). The *g*-factor value increased from 0.002 to about 0.008 when the concentrations of cysteine was increased from 0.4 to 2.0 μM , but it dropped to around 0.004 when the concentration of cysteine reached to 2.4 μM . Nevertheless, the symmetric *g*-factor curves in terms of L-Cys and D-Cys samples indicate the chiral characteristics of their corresponding plasmonic nanoparticles. According to the comment, relevant description about the CD spectra and *g*-factor results has been added in lines 142-157 (pages 5 and 6) in the revised manuscript. Fig. R5 and Fig. R6 are introduced in revised manuscript as Fig. 3e and Fig. 3f, respectively.

Fig. R5 Circular dichroism spectra of Au/MoS₂ nanostructures synthesized in the absence (control sample) and presence of diverse concentrations of L-Cys and D-Cys (0.4-2.4 μM), respectively.

Fig. R6 *g*-factor spectra of Au/MoS₂ nanostructures synthesized in the absence (control sample) and presence of diverse concentrations of L-Cys and D-Cys (0.4-2.4 μM), respectively.

Q7. The authors need to provide the difference between R- and S-dendritic nanocrystals in the demonstrated photothermal effect and drug release efficiency.

Response: According to the comment, we recorded the UV-vis-NIR absorption

spectra of *R*- and *S*-enantiomers of DNCs-1 and DNCs-2, it was found that the different conformation of nanostructures exhibit the consistent absorption performance (Fig. R7a). Nevertheless, DNCs-2 owned higher NIR absorption intensities than DNCs-1. None of differences in *R*- and *S*-enantiomers could be found in aspects of photothermal conversion performances (Fig. R7b). Additionally, compared to (*S*)-enantiomers of ofloxacin, (*R*)-enantiomers exhibit contrast impacts on levofloxacin. Larger loading values of nanostructures and smaller release efficiencies of nanostructure hydrogels could be calculated on (*S*)-enantiomers (Fig. R7c). Fig. R7 is shown in Fig. 5 in the revised manuscript, and relevant description has been added in lines 352-364 (pages 14 and 15).

Fig. R7 a UV-vis-NIR absorption spectra showing the differential optical absorption performances of CNCs, DNCs-1, and DNCs-2 hydrogels. **b** Temperature changes of aerogels and hydrogels doped with diverse kinds of nanostructures before and after the IR (808 nm) treatments. **c** Release ratios of levofloxacin and ofloxacin on the blank agarose hydrogel and Au/MoS₂ nanostructure doped agarose hydrogels after the IR treatments for 10 min. Hydrogels are pre-loaded with levofloxacin and ofloxacin (50.00 $\mu\text{g mL}^{-1}$), respectively.

Reviewer #3: This manuscript deals with the synthesis of chiral Au nanostructures using MoS₂ as templates and amino acids as their application for photothermal-response drug release. It is an interesting and relevant work, but the manuscript is very confused and difficult to follow. Besides, part of the discussion is not very well supported by experimental data I think that the authors must perform a deep revision of the manuscript. Therefore, my consideration is “not publish”.

We are sorry for our previous writing version, which might cause misconceptions to readers about our novelty and key points. According to all comments, we have revised our manuscript in academic writing. Additionally, supplementary experimental, theoretical studies, and the corresponding discussion have been added in the revised manuscript for a better understanding. Appreciate for your constructive comments.

Q1. Regarding the synthesis of the Au nanostructures in the presence of MoS₂ and in the presence or absence of amino acids. The discussion is based on the changes in the color and it is not very scientific. Based on the dependence of the optical properties in the size and shape of the Au nanoparticles, the authors to support their assumptions must perform the analysis of their optical properties using UV-Vis spectroscopy. Please, also revise the text, line 83, Figure 1a does not match there. Besides, the authors must include SEM/TEM analysis of the particles obtained in the absence of MoS₂.

Response: Both semiconducting MoS₂ layers and plasmonic gold nano-plasmonics exhibit characteristic absorption peaks, and in order to monitor the growth of Au nanostructures, the time-dependent Vis-NIR absorption spectra were recorded (Fig. R8a and b). The relevant description about the color changes have been replaced by the time-dependent optical analysis of reaction solutions, which is presented in lines 109-125 (pages 4 and 5) in the revised manuscript. We have revised the main text, pointed by Reviewer 3#, about Fig. 1a. Additionally, according to the comment, scanning electron microscope (SEM) experimental results indicating the morphologies CTAB-Au⁺ reaction product with the addition of exfoliated MoS₂ (Fig. R8c) and high-concentration L-Cys (Fig. R8d) were recorded, respectively. It was

found that the reaction time of CTAB-Au⁺ in the absence of MoS₂ largely extended. Meanwhile, the final products were unstable, which rapidly aggregated and turned to size-larger sediments. In the absence of MoS₂, we found the unprecedented morphologies of twisted nanowires in the samples of L-Cys involved gold nanostructures, rather than the dendritic nanostructures (Fig. R8d). Of significance, it is worthy to exploit the formation of twisted nanowires of plasmonics in our further works. Fig. R8 is presented in revised Supplementary Information (Supplementary Fig. 2). Thanks for your comments.

Fig. R8 **a** Time-resolved optical absorption spectra reflecting the reaction processes between exfoliated MoS₂ nanosheets and CTAB-Au⁺ in the absence and presence of 1.2 and 1.6 μM L-Cys. **b** Time-resolved optical absorption spectra of the CTAB-Au⁺ solution with the addition of 1.6 μM L-Cys. **c** and **d** SEM images of CTAB-Au⁺ reaction products with the addition of exfoliated MoS₂ (**c**) and 1.6 μM L-Cys (**d**), respectively.

Q2. The authors mention the importance of the size of MoS₂, nevertheless no information about that is included.

Response: In previous works, MoS₂ nanosheets were employed as both reductant reagents and stabilized ligands to directly react with HAuCl₄, and consequently, the Au/MoS₂ heterostructures were obtained. It was found that the diameters of Au/MoS₂ heterostructures were relied on the sizes of MoS₂ nanosheets.^{8,9} Nevertheless,

[8] Li, B. L., Zou, H. L., Luo, H. Q., Leong, D. T. & Li, N. B. Layered MoS₂ defect-driven in situ synthesis of plasmonic gold nanocrystals visualizes the planar size and interfacial diversity, *Nanoscale* **12**, 11979-11985 (2020).

[9] Li, B. L. *et al.* Principle of proximity: Plasmonic hot electrons motivate donator-adjacent

herein, our dendritic nanocrystals were obtained from the CTAB-Au⁺ reaction systems in the presence of MoS₂ and cysteine molecules, which were different with our previous works.^{8,9} We could not offer strong evidence to support the origin of size diversity. This academic issue could be addressed in our further studies. According to the comment, we deleted the description about the size influence of MoS₂ in the revised manuscript, highlighting the reaction mechanism of MoS₂ in directing the growth of Au plasmonics at the heterostructural interfaces. Relevant description is added in lines 207-223 (page 9), in the revised manuscript, and supplementary Fig. 3. Thanks for your constructive comments.

Q3. In line 87, the sentence “It was supposed that discriminated growths of nanostructures, induced by the small molecules of amino acids in the homogeneous solution, were activated” It is not supported by any data at this point.

Response: Sorry for our typos. In this sentence, the word “homogeneous solution” might cause misconceptions, which should be corrected to “aqueous solution”. In our original manuscript, we introduced the reactions between CTAB-Au⁺ and MoS₂ nanosheets in an aqueous solution. According to the comment, we revised this sentence in the revised manuscript. In the absence of MoS₂, L-Cys could modulate the growth of Au nanostructures and consequently, the aggregated Au wires could be obtained. Nevertheless, with the addition of MoS₂, the dendritic nanocrystals were formed because under the impact of MoS₂ interfaces, the transition state of highly faceted nano-seeds were involved. Relevant description has been added in lines 229-233 (page 9) in the revised manuscript.

Q4. In line 91, it is written that the particles are like a paper crane, from my point of view there is no need of including that. The particles are multibranched as showed the SEM and TEM analysis.

Response: The paper crane-related description and figure were deleted. Instead, we introduce the nanoarchitectures with the appearance of folded wings. Herein, the

semiconductor defects with enhanced electrocatalytic hydrogen evolution. *Nano Energy* **60**, 689-700 (2019).

folded wings are the key to understand the chiral structures. Thanks for your suggestion.

Q5. The CD is closely related to the absorption properties of the nanoparticles therefore for me the optical properties should be analyzed together and not afterward. Anyway, I do not agree with the analysis of the optical properties showed in lines 119-121, it is very poor. The same when analyzing the amount of L-cys. Besides, how is the g value obtained for these nanostructures? Besides, no discussion is performed about the effect of the amount of Cys.

Response: According to the suggestions, we analyzed the optical properties of as-synthesized nanomaterials in the beginning. We revised the analysis of optical absorption performance in the revised manuscript. Moreover, we measured the CD spectra and optical absorption spectra of nanocrystals aqueous dispersions, and the g-factor values were calculated based on the equation. It was found that the g-factor ranges from 0.002 to 0.008, which exhibited the similar regulation with previous works of nano-plasmonics. Significantly, the CD, SEM, optical absorption records were employed to study the effects of cysteine. It was found that the analysis results of SEM could consistently correspond to the regulation of CD and optical absorption spectra. The morphological modulation could be supported by the addition of cysteine, which was employed for the proposal of growing mechanism shown in Fig. 3 in the revised manuscript. The relevant description is added in lines 117-132 (pages 4 and 5), lines 146-157 (pages 6 and 7), and lines 208-219 (page 9) in the revised manuscript. Thanks for your comments.

Q6. The discussion about the changes in the morphology of the nanostructures is supported by TEM and SEM images. It is correct but the conclusions are overestimated. For me, it is not clear the slight zigzag that the authors observed on the branch of BNCs. Even I am not able to see the differences between L-Cys and D-Cys (2j and 2k). To claim that the authors should perform 3D tomography.

Response: This comment really helps us to improve this work a lot. According to the suggestion, Diverse-angle TEM images of nanostructures endowed by L-Cys and

D-Cys were presented (Fig. R9a, b), and we have conducted the 3D tomography tests for our synthesized Au nanostructures (Fig. R9c). The simulation results show that the folded branches appeared on the helicoids, and it was supposed that the molecular asymmetry of cysteine could result in the morphological asymmetric characteristics of the artificially synthesized nanostructures (Fig. R9d). In previous chiral works, because gold nanostructures were employed as the growth seeds, only plasmonic helicoids, rather than dendritic structures, were found. Herein, the chiral branches of dendritic nano-plasmonics inheriting from the amino acids are unprecedentedly synthesized, and we strongly proposed that this characteristic could be attributed to essential involvement of MoS₂ heterostructural surfaces, which was detailedly studied in this work. The analysis results were added in lines 178-187 (page 7) in the revised manuscript. Fig. R9 has been added in Fig. 2g-j in the revised manuscript. Moreover, the videos of 3D tomography data are shown in Supplementary information (Supplementary Movies 1, 2).

Fig. R9 a, b Diverse-angle TEM scan (a, b) and corresponding TEM tomography images (c) of individual nanostructures of Au/MoS₂ nanostructures synthesized in the presence of 1.6 μ M L-Cys (a) and D-Cys (b), respectively. d Asymmetric morphological simulation of synthesized Au/MoS₂ nanostructures derived from enantiomers of cysteine which is based on the TEM tomography images of nanostructures from L-Cys and D-Cys (c), respectively.

Q7. Regarding the use of other compounds such as BSA or GSH, it is not clear to me which is the aim of this work obtained branched nanoparticles or chiral nanoparticles with a high chiral response. Since no discussion about the chirality is included in the main manuscript. Moreover, for those experiments why was a concentration of 1.6 microM used?

Response: Appreciate for your comment. In our original manuscript, the works of BSA and GSH has none of relationship to the chiral nanostructures. We changed the description of BSA and GSH data, highlighting the role of BSA and GSH in directing the formation of branched nanostructures. The attachment of thiol groups on the highly faceted heterostructures results in the anisotropic growth of gold nanostructures. We will further study the information of thiol compounds, impacting the growth of plasmonic nanoparticles. The concentration of 1.6 micromolar was used for presenting the comparison between BSA, GSH, and cysteine. It was found that cysteine with the smallest molecular structure possessed the largest impact promoting the branches of Au/MoS₂ nano-heterostructures. According to the comment, we have added the description in lines 233-242 (pages 9 and 10) in the revised manuscript.

Q8. Regarding the mechanism, I suggest improving the discussion and performing a time-resolved TEM analysis to observe the evolution of the nanostructures during its formation.

Response: The time-resolved TEM analysis was conducted for tracking the morphological changes during the growth processes. The reaction was terminated via the ligand exchange and solvent transfer.^{10,11} The sediments were re-dispersed into ultrapure for further TEM characterizations. We have conducted the TEM records of Au/MoS₂ nanostructures in the absence (Fig. R10a) and presence of diverse

[10] Park, K. et al. Growth mechanism of gold nanorods. *Chem. Mater.* **25**, 555-563 (2013).

[11] Goulet, P. J. G., Bourret, G. R., Lennox, R. B. Facile phase transfer of large, water-soluble metal nanoparticles to nonpolar solvents. *Langmuir* **28**, 2909-2913 (2012).

concentrations of cysteine (Fig. R10b, c). Au/MoS₂ nanostructures can be arrested by the ligand exchange and solvent transfer. The time-resolved TEM images of Au/MoS₂ nanostructures were employed for exploiting the role of MoS₂. In terms of CNCs, the sizes of Au crystals increased from 50 to 120 nm, when the reaction time increases from 20 to 480 s (Fig. R10a). The cracks of CNCs were always accompanied by the thin nanosheets. In the presence of 1.2 μM L-Cys, the CNCs structures can be found within 120 s (Fig. R10b). HRTEM indicated that the thin layers near the nanocrystals were supposed to be the complete S-Mo-S hex-atomic rings and semiconducting-phase characteristics were identified. After 180 s, the branches start to be observed and finally turned to the dendritic nanoarchitectures with folded wings after 480 s. When the concentrations of L-Cys increased to 1.6 μM, the cracked morphologies could be only observed within 20 s. Furthermore, the branched structures were rapidly formed (Fig. R10c). The sizes of DNCs-2 gradually increased, and finally kept consistent after reaction for 480 s. According to the comment, relevant description has been added in lines 207-219 (page 9) in the revised manuscript.

Fig. R10 a-c TEM and HRTEM indicating the time-resolved morphologies of Au/MoS₂ CNCs (a), DNCs-1 (b), and DNCs-2 (c), respectively.

Q9. Regarding the discriminated growth, the structure of the resulting nanoparticle has not been analyzed in detail. Therefore, I do not understand why authors in line 174 claim that the particles are stellated octahedron differentiated by [321] facets. It is not supported by the experimental data. As mention before, it is difficult to believe for the data shown that BNCs have L-cys and D-cys structures (fig J and K).

Response: Thanks for your comment. In order to uncover the structure of dendritic nanocrystals, time-resolved TEM, 3D tomography tests were conducted. We analyzed the growth mechanism of Au/MoS₂ nanostructures in the absence of presence of cysteine. From the time-resolved TEM, it was found that the nanoseeds were grown on the MoS₂ heterostructural interfaces, and highly faceted polyhedrons were obtained. According to previous works, the arrangement of crystal surfaces was differentiated by {321} facets. Due to the discriminated interaction effects of cysteine enantiomers to {321} facets,¹² the anisotropic growth of Au nanostructures was followed with the merging chiral characteristics. Moreover, the 3D tomography results identify the differential twisted direction of branches, contributing to the appears of symmetric morphologies in nanomaterials. The relevant description is shown in lines 178-187 (page 7), lines 207-219 (page 9), lines 256-264 (page 10) in the revised manuscript.

Q10. Regarding the experiments of the nanostructures for photothermal-response drug release, I do not know the meaning of the sentence of Line 225-227 “The differential conformation of BNCs and DNCs exhibit the consistent absorption performances (supplementary Fig 7)”. Besides, why they selected the DNCs nanoparticles for the photothermal experiments (figure 3c) and not the other chiral nanoparticles obtained.

Response: We identified that the enantiomers of DNCs-1 and DNCs-2 exhibited the consistent optical absorption and photothermal performances. The results are important to show that the discriminated effects in drug release are originated from the different interaction between ofloxacin and enantiomers of Au/MoS₂ dendritic nanocrystals. The DNCs-2, previously named as DNCs, were used for photothermal studies because they own the comparable IR (808 nm) absorption performance and highest photothermal conversion efficiencies. The comparison in IR absorption and photothermal studies is shown in lines 326-331 (page 13), and Fig. 5b and c, in the revised manuscript.

[12] Lee, H. E. et al. Amino-acid- and peptide-directed synthesis of chiral plasmonic gold nanoparticles, *Nature* **556**, 360-365 (2018).

Q11. In figure 4f it is showed the loading efficiency of the hydrogels, right? Then I do not know why in the discussion it is explained as release ratios (lines 268-270). The discussion about the differences in the release among the different nanoparticles is very confusing and the message is not clear.

Response: Thanks for your comments. We have revised the description of Fig. 5f in the original manuscript. In order to assess the loading efficiency, nanostructures were added into the levofloxacin, and the mixture was incubated for 2 h. Subsequently, the mixture was centrifuged at 10000 rpm for 10 min, and nanostructures settle down. The concentrations of residual levofloxacin were measured for calculation of loading efficiencies. It was found that the in terms of DNCs-1 and DNCs-2, the loading efficiencies of (S)-enantiomers are higher than that of (R)-enantiomers, which could be attributed to the increased absorption interaction between levofloxacin molecules and (R)-enantiomers. Additionally, DNCs-2 exhibited the larger loading efficiency than DNCs-1. Considering that the dendritic degrees in DNCs-2 enhanced, the larger surface-to-volume ratios resulted in an increasing amount of levofloxacin on surfaces of nanostructures. Overall, due to the comparable photothermal performances and drug-loading efficiencies, DNCs-2 with different conformation was employed for the studies of advanced drug release systems. The in-vitro experiments were conducted, showing that the levofloxacin could release from the nanostructure-doped hydrogels to the aqueous solution. When the IR treatments were applied, the release ratios in drug-loading DNCs-2 hydrogels largely increased. Compared to the room-temperature, the release ratios increased almost three times in terms of IR treatments. According to the comment, relevant description has been added in lines 338-346 (page 14) in the revised manuscript.

Reviewer #4: The manuscript “Dendritic nanocrystals inheriting the chirality from amino acids at heterostructural interfaces” by the anonymous authors is on the very topical subject of chirality transfer.

Thanks for your revision suggestions. Sorry for our previous weak English writing skills, and we have significantly revised our manuscript in both introduction and experiment description. On the basis of our efforts, it was suggested that the revised version would reach the high standard of *Nat. Commun.* for publication. Thanks.

Q1. The introduction lacks structure. The general background is not presented clearly. There is no clearly defined problem and the question(s) the authors are trying to answer is not presented. The key concepts are not introduced (explain: what is chirality? What is chirality transfer? What is quantum confinement-determined optical activity? Why are amino acids important/essential? When talking about “chiral recognition capacity”, who is recognizing what and why should the reader know about it? What are metamaterials?) The state of the current literature is not adequately surveyed – this is a paper about chirality transfer, there should be many references on chirality transfer with clear explanations: e.g. “such and such demonstrated chirality transfer from this to that”. Instead, there is a collection of vaguely related sentences and references, generally on the topic of chirality and some references on chirality transfer but without clear explanations or a unifying narrative.

Response: We find that this suggestion is really important. Supplementary experimental results have been conducted and the description about the chiral transferring was correspondingly re-wrote. Of significance, after the revision, the logic of our protocol is better understood. In addition, we have also re-organized our introduction on the basis of above suggestions.

Q2. The manuscript it not written in clear English, e.g. “Nevertheless, their potential performances are unfortunately inactivated.”

Response: Sorry for our English skills. According to the comment, we have improved our English writing in the revised manuscript. We hoped that our revision could reach the high standard for publication.

Q3. The interaction between cysteine and MoS₂ is intriguing and potentially interesting. The way the data are presented though is unclear. What is this origami image? Why is it there? All acronyms should be defined. In figure 1, are the CD spectra reproducible or have they been hand-picked to show opposite CD? Are the CD spectra from single nanoparticles or from the whole sample?

Response: According to the comments, we deleted the origami image. All acronyms have been well defined. Thanks for this point. The CD spectra were reproducible. In order to address other reviewer's concerns, we have re-measured our samples with their higher concentrations. It was found that in addition to the intensity, the peaks shown in CD spectra of the revised manuscript exhibited consistent regulation which is only dependent on the concentrations of cysteine. Significantly, the mirror-symmetric negative/positive peaks could be found on the samples of L-Cys and D-Cys with the same concentrations. The CD spectra were detected in the aqueous nanoparticle dispersions. In order to eliminate this concern, we have added these points in the revised manuscript, shown in lines 142-157 (pages 5 and 6).

Q4. Figure 2 is very interesting and could almost be a paper by itself, demonstrating that these are separate stages indeed, and what the conditions for each case are.

Response: We divided the data of Fig. 2 in the previous manuscript in Fig. 1 and Fig. 3 in the revised manuscript. Firstly, the optical spectra and SEM images identified the morphological diversity of Au/MoS₂ nanoarchitectures, uncovering the branched and dendritic structures of Au/MoS₂. Additionally, the TEM and HRTEM images of Au/MoS₂ nanoarchitectures synthesized in the absence of cysteine indicating the highly faceted nanocrystals could be formed with the involvement of MoS₂ heterostructural surfaces. Based on the time-resolved TEM records, the anisotropic growth mechanism of dendritic nanocrystals with the modulation of MoS₂ heterostructural interfaces and enantiomers of cysteine was proposed. Currently, as our most important novelty, we explain the reaction mechanism from the data presented in Fig. 2 of the original manuscript. Moreover, we will focus on the exploitation in changes of optical spectra and SEM morphological diversity in our next

works. Thanks for your suggestions.

Q5. Figure 3 is very unclear and the caption does not help much to understand what is what. Again, if sufficiently explained and backed up by scientific evidence at every statement, this might be a paper by itself. Where is the cysteine?

Response: In previous Figure 3, we aim at developing the comparable applications of as-synthesized Au/MoS₂ heterostructures. Without the surface functional protection, nanomaterials own weak stabilities in aqueous solution. Herein, in order to achieve the high application values, an intriguing substrate should be integrated with the characteristics of Au/MoS₂ heterostructures. Considering those attractive performance of hydrogels, we used the agarose hydrogel as the working substrate of as-synthesized nanomaterials, and on the basis of enantiomer-dependent interaction of nanomaterials with chiral molecules. The responsive and sustainable drug release system was established based on the IR-heat transformation, morphological structures of nanomaterial-doped hydrogels. In this part, DNCs were prepared from the contribution of cysteine, which exhibited different enantiomers relied on the L-Cys and D-Cys. Herein, the (*S*)-DNCs refer to the Au/MoS₂ heterostructures endowed by L-Cys, whereas (*R*)-DNCs correspond to the D-Cys. According to the comment, we have explained the definition in our revised manuscript, shown in lines 256-264 (page 10).

Q6. Figure 4 could likely also be a paper by itself, with more data to actually show that this is happening as the authors say it does. The same comment for Figure 5.

Response: In previous Figure 4 (Fig. 5 in the revised manuscript), theoretical simulation indicated that the chiral crystal faces exhibited differential interaction to (*S*)-ofloxacin and (*R*)-ofloxacin, supporting the developed chiral release protocol. Next, we went further into exploiting the differential chiral performance via the theoretical simulation and experiments. Additionally, Fig. 5 (Fig. 6 in the revised manuscript) shows the antimicrobial applications in IR-responsive and sustainable performance of chiral Au/MoS₂ nanoarchitecture hydrogels. This work definitely guides the potential of our nanoparticles with enantioselective applications.

Q7. I assume that the authors have actually read reference 2 (and all the references they cite). Does this paper really illustrate chirality transfer or does it say that there is no chirality transfer in the Te systems?

Response: The tellurium nanocrystals were synthesized on the basis of bottom-up methods. In the presence of chiral thiolated penicillamine ligands, the reduction of tellurium dioxide results in the synthesis of chiral nanostructures.¹³ The growing processes were uncovered. It was found that thin twisted nanorods were formed, and after experiencing the formats of trigonal bipyramids, polyhedron possessing a D3 point symmetry was obtained. It indicates the contribution of chiral thiolated ligands. Of significance, similar with the chiral plasmonic nanoparticles, chiral transfer from the small molecules to the nanostructures was proposed. According to the comment, we added the description about the introduction of the published tellurium nanocrystals system in the revised manuscript, shown in lines 41-44 (page 2).

Q8. Overall, this is an intriguing work and it might well be interesting to a broad community of readers. Unfortunately, the presentation is poor to the point of incomprehensible. The lack of clarity in English results in a lack of rigor, so some of the scientific claims seem unsubstantiated. I also suspect that the authors are trying to put too much work into a single paper, without satisfying the requirements for sufficient detail. The authors might be well advised to split this work in 3 to 5 separate but related manuscripts, making sure that every single scientific claim is supported by direct evidence.

Response: Thanks for your kind comments. We have re-arranged our manuscript, and highlighted our novelty in the revised manuscript. The experiments were conducted around the formation of dendritic nanocrystals, chiral characteristics including morphologies and molecular interactions, as well as chirality-dependent IR responsive and sustainable molecule release systems. This work is important to

[13] Ben-Moshe, A. *et al.* The chain of chirality transfer in tellurium nanocrystals. *Science* **372**, 729-733 (2021).

promote the studies of unprecedented chiral dendritic nanostructures with heterostructural composition. According to the comment, we re-organized the experimental data and correspondingly re-wrote the manuscript for the improved academic logic and high-level quality.

Reviewer comments, second review

Reviewer #1 (Remarks to the Author):

I appreciate the authors' efforts in clarifying their claims, and this is a significantly revised manuscript with more data. However, I am still concerned that these data are not enough to support all the conclusions in the manuscript strongly, and the novelty and importance of this work are not fit with the standard of Nature Communications.

1, The title highlights that chirality transfers from amino acids to dendritic nanocrystals. However, the data is not strongly supporting the transfer mechanism. In this work, the growth mechanism and process were well characterized, but how the chirality transfer from amino acids to nanocrystal is unclear. Also, how the cysteine interacts with the MoS₂ seed after pre-coating? Whether it is possible that the cysteine may interact with the early gold nanostructures after it grows on MoS₂ and then transfers the chirality to later dendritic nanocrystals rather than cysteine coated on the MoS₂? What will happen if you remove the amino acid from the growth solution after incubating with the MoS₂ and repeat the growth process? Whether the chirality still can be transferred into the dendritic nanocrystals?

2, The photothermal-responsive drug release is a good drug-controlled delivery and release strategy that previous works have widely validated. But it is important to ask whether using the dendritic nanocrystals with chirality in hydrogels for drug release is necessary (what is the advantage of presented dendritic nanocrystals for drug release?) and whether the chirality plays a vital role in the loading efficiency, release ratio, and antimicrobial characteristics. From my point of view, although the effects are weakly different between (S)-DNCs-2 and (D)-DNCs-2, the advantage and contribution of chirality to the loading efficiency, release ratio, and antimicrobial characteristics are minimal compared with the published works. Maybe the dendritic nanocrystals with chirality properties can be used for other purposes.

3, The author concluded in line 481 that the differential interaction effect of enantioselective dendritic nanoarchitectures to levofloxacin contributed to the modulated drug release performance. Still, there is no data to support this claim. As shown in Figure 5i, ofloxacin and levofloxacin have almost the same release ratio (the significant differences should be added between levofloxacin and ofloxacin) in (S)-DNCs-2 or (D)-DNCs-2, so it is hard to say that the dendritic nanoarchitectures has chiral recognition and enantioselective capacity. Further, the release mechanism of loaded levofloxacin is unclear again. Whether the interaction between levofloxacin molecules and the dendritic nanoarchitectures control the release, or just because the photothermal effect accelerated molecular diffusion (levofloxacin molecules just loaded in the hydrogels but no interaction with dendritic nanoarchitectures)? To better understand the release mechanism, it is important to study the interaction relationship between levofloxacin molecules and the dendritic nanoarchitectures.

Reviewer #2 (Remarks to the Author):

The author did a lot of efforts including the new experiments and the discussion to address the concerns by four reviewers. I think that scientific questions are fully and correctly answered. Thus, I recommend the publication in Nature Comm.

Reviewer #3 (Remarks to the Author):

The authors have made a great effort to improve the quality of the manuscript, showing additional data and discussion. My consideration is accept it.

Point-to-Point Responses to Reviewers' Comments (NCOMMS-22-07182A-Z)

We would like to express our gratitude to reviewers for three reviewers' encouraging and constructive comments. All changes in the revised manuscript are marked with red font for tracking purposes.

Reviewer #1: *I appreciate the authors' efforts in clarifying their claims, and this is a significantly revised manuscript with more data. However, I am still concerned that these data are not enough to support all the conclusions in the manuscript strongly, and the novelty and importance of this work are not fit with the standard of Nature Communications.*

Thanks for your constructive comments. We have revised our manuscript, and in order to reach the high standard of *Nat. Commun.*, we provided supplementary data for supporting the conclusion of our work. In addition, the description of our manuscript was re-organized, aiming at highlighting our novelty of the chiral transfer of amino acids to plasmonic gold nanostructures at hetero-structural MoS₂ surfaces. Overall, the point-to-point revision list and responses are shown below:

Q1. *The title highlights that chirality transfers from amino acids to dendritic nanocrystals. However, the data is not strongly supporting the transfer mechanism. In this work, the growth mechanism and process were well characterized, but how the chirality transfer from amino acids to nanocrystal is unclear. Also, how the cysteine interacts with the MoS₂ seed after pre-coating? Whether it is possible that the cysteine may interact with the early gold nanostructures after it grows on MoS₂ and then transfers the chirality to later dendritic nanocrystals rather than cysteine coated on the MoS₂? What will happen if you remove the amino acid from the growth solution after incubating with the MoS₂ and repeat the growth process? Whether the chirality still can be transferred into the dendritic nanocrystals?*

In order to clearly respond to this comment, reproduced verbatim, we divided this comment into four questions, and point-to-point responses are shown below.

Q1-a: The title highlights that chirality transfers from amino acids to dendritic nanocrystals.

However, the data is not strongly supporting the transfer mechanism. In this work, the growth mechanism and process were well characterized, but how the chirality transfer from amino acids to nanocrystal is unclear.

Our response: Thank Reviewer #1 for your questions and your kind comments that the growth mechanism and process were well characterized. The chirality transfer is our way of describing the handedness aspects from the chiral template to the nanocrystal. It is like to a chiral imprint on a stamp dipped into ink and then stamped onto paper. There is no real transfer of the materials on the stamp but the chiral properties are transferred from the stamp to paper. We know this is not the best analogy but it suffices the various conceptual elements that we are trying to portray through the term “chirality transfer”. Our key point is the formation of highly-faceted Au/MoS₂ nanocrystals, which served as intermediate states for the attachment of cysteine enantiomers. After that, the cysteine molecules play an important role in guiding the sequential growth of plasmonic nanostructures and chirality characteristics could be found on these products of dendritic nanostructures. Our proposed mechanism about the chirality origin from the immediate Au/MoS₂ hybrids is accordance to the work published before¹. We also cited and introduced this work in our manuscript. According to the comment, supportive experiments shown in Q1-b and Q1-d were conducted to identify our view and we correspondingly added relevant description about the chirality transfer mechanism in lines 245-258 (page 10) in the revised manuscript.

Q1-b: Also, how the cysteine interacts with the MoS₂ seed after pre-coating?

Our response: The interaction between MoS₂ and cysteine is based on the physisorption effect ². We also found that compared to proteins, cysteine, as a kind

[1] Lee, H. E. et al. Amino-acid- and peptide-directed synthesis of chiral plasmonic gold nanoparticles. *Nature* **556**, 361 (2018).

[2] Chen, X., Berner, N. C., Backes, C., Duesberg, G. S. & McDonald, A. R. Functionalization of two-dimensional MoS₂: On the reaction between MoS₂ and organic thiols. *Angew. Chem. Ed. Int.* **128**, 5897-5902 (2016).

of small molecules, exhibited weaker interaction forces to MoS₂ surfaces and its adsorption cannot affect the growth of Au nanostructures on MoS₂³. In order to prepare materials with high reproducibility, we simplified our synthesis routes, adding cysteine and MoS₂ together in the growth solution, and the operation process was optimized. Of significance, the growth seed solution was present with the addition of cysteine in MoS₂ solution without any incubation treatments. After that, the MoS₂-cysteine mixture was added into the CTAB-Au⁺ growth solution for the reactions. In order to address the concerns of reviewer, extra experimental results are provided to identify that the cysteine in the solution, instead of those adsorbed on MoS₂ surfaces, can induce the eruption of chirality in hybrid nanostructures. Consequently, we designed the corresponding experiments. Firstly, exfoliated MoS₂ was added to the growth solution (CTAB-Au⁺). After different reaction times (20 s to 10 min), the cysteine solutions containing different L/D-enantiomers were added to the reaction solutions, respectively, and circular dichroism (CD) spectra were recorded to characterize the final products. When the reaction times increased from 20 s to 1 min, the CD curves remain constant but their peaks slightly shift to longer wavelengths. However, the CD peaks gradually drop after the reaction times increase to 1 min, and none of CD signals are observed after 5 min. The chiral nano-heterostructures were synthesized based on the MoS₂ interfaces. After the MoS₂ nanosheets were totally covered by the Au atomic layers, the products could not be served as the growth seeds for the sequential formation of chiral Au/MoS₂ nano-heterostructures. The relevant description has been added in lines 252-258 (page 10) in the revised manuscript. Fig. R1 was added in the revised Supporting Information (Supplementary Fig. 7b).

[3] Li, B. L. et al. Layered MoS₂ defect-driven in situ synthesis of plasmonic gold nanocrystals visualizes the planar size and interfacial diversity. *Nanoscale* **12**, 11979 (2020).

Fig. R1 Circular dichroism spectra of as-synthesized Au/MoS₂ hybrids in the presence of L-Cys and D-Cys, respectively. The addition of 1.2 μM cysteine (L-Cys or D-Cys) was conducted after the exfoliated MoS₂ was added in the growth solutions (CTAB-Au⁺) for different reaction times (20 s, 1 min, 3 min, 5 min, 10 min). As control samples, L-Cys and D-Cys were mixed with MoS₂, and then the mixtures were added to the growth solutions (L-Cys/MoS₂ and D-Cys/MoS₂).

Q1-c: Whether it is possible that the cysteine may interact with the early gold nanostructures after it grows on MoS₂ and then transfers the chirality to later dendritic nanocrystals rather than cysteine coated on the MoS₂?

Our response: Actually, our mechanism in the original manuscript is exactly consistent to the conception of reviewer. The chirality characteristics of nanomaterials cannot be found in the beginning even the seed solution containing MoS₂ and 1.2 μM cysteine was added (Fig. 3b in the revised manuscript). The chirality transfer mechanism is not due to the adsorption of cysteine on MoS₂ surfaces. Alternatively, the addition of MoS₂ in the growth solution firstly results in the construction of highly-faceted nanocrystals. Subsequently, the cysteine modulates the eruption of chirality for the formation of chiral Au/MoS₂

nano-heterostructures based on the significant theory mentioned by Kim *et al.*¹. We are sorry that our mechanism about the formation of chiral nano-heterostructures was not clearly present and it caused some unintentional misunderstanding. We enhanced the description of our growth mechanism in the revised manuscript, shown in lines 254-258 (page 10) in the revised manuscript.

Q1-d: What will happen if you remove the amino acid from the growth solution after incubating with the MoS₂ and repeat the growth process? Whether the chirality still can be transferred into the dendritic nanocrystals?

Our response: The chirality of dendritic nanocrystals originates from the discriminated interaction of amino acids enantiomers on substrates. From the simulation computation, we cannot find any interaction differences between MoS₂ surfaces and cysteine enantiomers. In contrast, the {321}^S and {321}^R exhibited discriminated interaction to L-/D-cysteine, respectively. The highly faceted nanostructures of Au/MoS₂ possessed abundant chiral crystal facets for the enantiomer-dependent attachment of cysteine, and the sequential growth of Au nanostructures results in the formation of dendritic nanostructures. In order to respond the Q1-b, we conducted the experiments, sequentially adding the MoS₂ and cysteine to the growth solution. The experimental results indicate that the intermediate state of Au/MoS₂ guided the growth of chiral dendritic nanocrystals, instead of the MoS₂ layers. Reviewer #1 gave us a good suggestion. After the incubation of cysteine with MoS₂, the samples were centrifuged and the sediments were collected. We used the sediments as the seed materials for the growth of Au/MoS₂ hybrids. It was found that no CD responses are observed from the treated MoS₂ samples. CD spectra indicate for us that after the removal of free cysteine in solution, even small number of L-/D-cysteine molecules physisorbed on MoS₂ did not contribute to the growth of chiral materials. Appreciate the reviewer for the comment that resulted in greater clarity of the growth mechanism. Fig. R2 was added in the revised manuscript (Supplementary Fig. S7a). The experimental results have been added in lines 245-252 (page 10) in the revised manuscript.

Fig. R2 CD spectra of as-synthesized Au/MoS₂ hybrids before and after the free cysteine molecules (L-Cys and D-Cys) in the solutions have been removed via the centrifugation. The concentrations of L-Cys and D-Cys are 1.2 μ M, respectively.

Q2. The photothermal-responsive drug release is a good drug-controlled delivery and release strategy that previous works have widely validated. But it is important to ask whether using the dendritic nanocrystals with chirality in hydrogels for drug release is necessary (what is the advantage of presented dendritic nanocrystals for drug release?) and whether the chirality plays a vital role in the loading efficiency, release ratio, and antimicrobial characteristics. From my point of view, although the effects are weakly different between (S)-DNCs-2 and (D)-DNCs-2, the advantage and contribution of chirality to the loading efficiency, release ratio, and antimicrobial characteristics are minimal compared with the published works. Maybe the dendritic nanocrystals with chirality properties can be used for other purposes.

Our response: In this manuscript, we totally agree with both your points that photothermal responsive drug release is a well reported route and highly efficacious way to release the drug. On top of that, the release efficacy will be much higher since dendritic heterostructures can achieve very high photothermal temperatures irrespective on their handedness. However, that is like a brute force release method

and to kick the photothermal dendritic nanoparticles into overdrive would literally melt the gold dendrites. That would be equivalent to just burst release. There is also additional collateral damage to the drug that is released as well as the surrounding tissue in the body due to the excessive heat generated. Therefore, there are certain advantages to keeping temperatures on the cooler side while still maintaining some selectivity. Additionally, the chiral based relationship between the chiral drug and the chiral dendritic nanoparticles provides that selectivity without adding in exogenous agents (like antibodies) that complicates the design. The minimal inevitable heat is needed to increase entropy of the system that releases the drug but not so high as to cause other undesirable outcomes. We agree that much optimization needs to be done so our chirality driven strategy can be competitive with conventional pure photothermal driven release but the latter did have quite a few years of headstart. This weak performance is nonetheless a promising start in the right direction. Even photothermal performances of nanomaterials have been considered for the contribution of responsive drug release platforms, the release modulation of chiral drug molecules was rarely reported. The intrinsic properties of plasmonic photothermal behavior was integrated with the enantiomer-dependent interaction between plasmonic nanostructures and organic molecules. Of significance, the theory supporting the enantiomeric-dependent differences in drug release and antimicrobial assessment is still scarce. Our work can inspire other works in understanding the chirality origin, and differentiated enantiomeric performances. Thanks for your suggestions, we will further seek to under the mechanism involved in these chiral parameters of plasmonic nanocrystals and used them to improve on their biomedical applications. The high potential in construction of intelligent biomedicine platforms could be found. Thanks for your question.

A series of photothermal research, based on MoS₂ and plasmonic nanostructures, have been studied for cancer therapies. Additionally, the hydrogel without the modification of nanostructures, lacks the intriguing properties used for stimuli-responsive applications. Thus, the supplementary function could be

extended with the addition of nanostructures ⁴. We focus on the synthesis and applications of novel chiral plasmonic nanomaterials with dendritic structures. The photothermal-responsive applications in drug release indicate the potential applications of these unprecedented nanomaterials with chiral characteristics, which could exhibit highly promising roles in wide fields. For example, the performance can be improved based on the screen of substrates. Additionally, these new characteristics could inspire the application of emerging chiral plasmonic nanomaterials in construction of optoelectronic devices and understandings of important biological information, such as immunity^{5,6}. Furthermore, unknown aspects can be discovered based on the synthesis of chiral nanostructures and their synergistic performances with developed materials and technologies. We will go further studies in digging out the new properties of chiral dendritic nanostructures and achieving the improved applications. According to the comments, we added relevant description in the Conclusion part, shown in lines 522-526 (page 21) in the revised manuscript.

Q3. The author concluded in line 481 that the differential interaction effect of enantioselective dendritic nanoarchitectures to levofloxacin contributed to the modulated drug release performance. Still, there is no data to support this claim. As shown in Figure 5i, ofloxacin and levofloxacin have almost the same release ratio (the significant differences should be added between levofloxacin and ofloxacin) in (S)-DNCs-2 or (D)-DNCs-2, so it is hard to say that the dendritic nanoarchitectures has chiral recognition and enantioselective capacity. Further, the release mechanism of loaded levofloxacin is unclear again. Whether the interaction between

[4] Zhang, K. et al. Extravascular gelation shrinkage-derived internal stress enables tumor starvation therapy with suppressed metastasis and recurrence. *Nat. Commun.* **10**, 5380 (2019).

[5] Cai, J. R. et al. Polarization-sensitive optoionic membranes from chiral plasmonic nanoparticles. *Nat. Nanotechnol.* **17**, 408-416 (2022).

[6] Xu, L. G. et al. Enantiomers-dependent immunological response to chiral nanoparticles. *Nature* **601**, 366-373 (2022).

levofloxacin molecules and the dendritic nanoarchitectures control the release, or just because the photothermal effect accelerated molecular diffusion (levofloxacin molecules just loaded in the hydrogels but no interaction with dendritic nanoarchitectures)? To better understand the release mechanism, it is important to study the interaction relationship between levofloxacin molecules and the dendritic nanoarchitectures.

In order to clearly respond to this comment, reproduced verbatim, we divided this comment into two questions, and point-to-point responses are shown below.

Q3-a: The author concluded in line 481 that the differential interaction effect of enantioselective dendritic nanoarchitectures to levofloxacin contributed to the modulated drug release performance. Still, there is no data to support this claim. As shown in Figure 5i, ofloxacin and levofloxacin have almost the same release ratio (the significant differences should be added between levofloxacin and ofloxacin) in (S)-DNCs-2 or (D)-DNCs-2, so it is hard to say that the dendritic nanoarchitectures has chiral recognition and enantioselective capacity.

Our response: Appreciate for your suggestion. The significant result differences between levofloxacin and ofloxacin have been added in Fig. 5 and its figure caption. Our simulation results also corroborated with the bench data indicating the origins of enantioselective capacity of gold crystal to drug enantiomers. Currently, there are only slight differences of release ratios between levofloxacin and ofloxacin in our experiments is because diffusion of the released levofloxacin and ofloxacin is slowed down by the porous nature of the hydrogels. It was slowed down so significantly that the significant differences in release rates are partially negated but it still manages to show a statistically significant difference (please see our further responses to this similar concern in the ensuing comments). In our work, we focus on the synthesis of materials, and the functional properties of materials are primarily studied, especially in the field of chiral recognition. Furthermore, in order to facilitate the chirality-dependent biomedical applications, we should work on the optimization of substrates supporting the function of chiral plasmonic materials. In order to address the concern of reviewer, we conducted the interaction studies which are

shown in Q3-b. The assessment of our hydrogel-based release systems has been added in lines 391-393 (page 16) in the revised manuscript. Thanks for your comments.

Q3-b: Further, the release mechanism of loaded levofloxacin is unclear again. Whether the interaction between levofloxacin molecules and the dendritic nanoarchitectures control the release, or just because the photothermal effect accelerated molecular diffusion (levofloxacin molecules just loaded in the hydrogels but no interaction with dendritic nanoarchitectures)? To better understand the release mechanism, it is important to study the interaction relationship between levofloxacin molecules and the dendritic nanoarchitectures.

Our response: In order to briefly study the chirality characteristics of nanostructures, we conducted the chirality researches in the aqueous solution. Firstly, we incubated the levofloxacin and ofloxacin together with (S)-DNCs-2 or (R)-DNCs-2, respectively, for different times. Subsequently, the systems are centrifuged and the fluorescence intensities of levofloxacin and ofloxacin in the supernatant were measured (**Fig. R3**). The fluorescence intensities decrease with the increasing incubation times, indicating the rapid adsorption of drug molecules on plasmonic nanomaterials. More importantly, in (S)-DNCs-2 samples, the fluorescence changes of ofloxacin are larger than those of levofloxacin. In contrast, (R)-DNCs-2 exhibited higher fluorescence changes to levofloxacin. The contrast adsorption performances of (S)-DNCs-2 and (R)-DNCs-2 to ofloxacin enantiomers are identified. The experimental results of interaction studies are accordance to our previous experiment. We found that the saturated loading efficiency of levofloxacin on (R)-DNCs was higher than that on (S)-DNCs. As a control, no differences could be found on the CNCs, which are prepared in the absence of cysteine. In addition, from the time-resolved fluorescence records, we find that, the loading of drug molecules on DNCs is rapid, and the adsorption processes of ofloxacin are completed within 30 min. According to the comment, we added the experiment results and relevant description about the interaction between molecules and dendritic nanocrystals in

lines 353-366 (pages 14 and 15) in the revised manuscript. Fig. R3 was added in the Supporting Information (Supplementary Fig. 10) part of the revised manuscript.

The adsorption of levofloxacin on dendritic nanocrystals is based on the Van der Waals interaction. From the *in-vitro* experiment results (Fig. 5h in the manuscript), the release performance of drug is resulted by the heat-induced molecular motion. In comparison to the hydrogel samples without the modification of dendritic nanocrystals, the DNCs hydrogels exhibit improved sustainable release performance. The interaction of levofloxacin on dendritic nanocrystals in the hydrogels is identified. Additionally, the interaction between the drug molecules and dendritic nanoarchitectures contribute to the chirality-reliable sustainable release systems. In comparison to the hydrogel samples without the DNCs, the DNCs hydrogels exhibited IR-responsive characteristics, suggesting that the heat could accelerate the diffusion of molecular drugs, and consequently exogenous IR-responsive drug release systems were developed. Relevant description about the photothermal-induced drug release has been added in lines 377-380 (page 15) in the revised manuscript.

Fig. R3 Normalized fluorescence intensities in the supernatant responding to the various incubation times of (S)-DNCs-2, (R)-DNCs-2, and CNCs Au/MoS₂ nanostructures to ofloxacin racemate and levofloxacin, respectively. The concentrations of levofloxacin and ofloxacin are 10 $\mu\text{g mL}^{-1}$. The concentrations of nanostructures are 100 $\mu\text{g mL}^{-1}$. Data are presented as mean values \pm SD, $P < 0.05$.

Reviewer #2: *The author did a lot of efforts including the new experiments and the discussion to address the concerns by four reviewers. I think that scientific questions are fully and correctly answered. Thus, I recommend the publication in Nature Comm.*

Our response: Thank you very much for your valuable points and precious effort on improving the clarity of our manuscript. We appreciate the reviewer's supportive consideration for its publication on *Nature Communications*.

Reviewer #3: *The authors have made a great effort to improve the quality of the manuscript, showing additional data and discussion. My consideration is accept it.*

Our response: We thank the reviewer for the acknowledgment on the quality of our work. We appreciate for your agreement and recommendation of its publication.

Reviewer comments, third review

Reviewer #1 (Remarks to the Author):

Since the authors have corrected the revision as the reviewers mentioned, the current manuscript could be accepted and published without further modifications.

Point-by-Point Responses to Reviewer's Comments
(NCOMMS-22-07182C)

Reviewer #1: *Since the authors have corrected the revision as the reviewers mentioned, the current manuscript could be accepted and published without further modifications.*

Our response: Thank you very much for your valuable comments so that the clarity of our manuscript can be largely improved. We also appreciate the reviewer's supportive consideration for its publication on *Nature Communications*.